



# A continental reconstruction of hydroclimatic variability in South America during the past 2000 years

Mathurin A. Choblet[1,2,3], Janica C. Bühler[3], Valdir F. Novello[3], Nathan J. Steiger[4,5], and Kira Rehfeld[3]

[1]Department of Astrophysics, Geophysics and Oceanography, University of Liège, Liège, Belgium
[2]Institute of Environmental Physics, Heidelberg University, Heidelberg, Germany
[3]Department of Geosciences, University of Tübingen, Tübingen, Germany
[4]Hebrew University of Jerusalem, Jerusalem, Israel
[5]Lamont-Doherty Earth Observatory, Columbia University, Palisades, NY, USA

**Correspondence:** Mathurin A. Choblet (mathurin@choblet.com)

**Abstract.** Paleoclimatological field reconstructions are valuable for understanding hydroclimatic variability. While being similarly impactful on societies as temperature variability, hydroclimatic variability has still remained less in focus. However, reconstructing globally complete fields of climate variables lacks adequate proxy data from tropical regions like South America, limiting our understanding of past hydroclimatic changes in these areas. This study addresses this gap using low resolution

climate archives, including speleothems, previously omitted from reconstructions. Speleothems record climate variations on decadal to centennial time scales and provide a rich dataset for the otherwise proxy data scarce region of tropical South America. By employing a multi-time scale Paleoclimate Data Assimilation approach, we synthesize climate proxy records and climate model simulations, capable of simulating water isotopologues in the atmosphere, to reconstruct 2000 years of South American climate. This includes surface air temperature, precipitation amount, drought index, isotopic composition of pre-

cipitation amount, and the intensity of the South American Summer Monsoon. The reconstruction reveals anomalous climate periods: a wetter and colder phase during the Little Ice Age (∼1500 - 1850 CE) and a drier, warmer period corresponding to the early Medieval Climate Anomaly (∼600 - 900 CE). However, these patterns are not uniform across the continent, with exceptions in northeastern Brazil and the Southern Cone, indicating regional variability. The anomalies are more pronounced than in previous reconstructions, but align with local proxy record studies, thus highlighting the importance of including speleothem

proxies. The multi-timescale approach is essential for reconstructing multi-decadal and centennial climate variability. Despite methodological uncertainties regarding climate model biases and proxy record interpretations, this study marks a crucial first step in incorporating speleothems into climate field reconstructions, potentially enhancing insights into past hydroclimatic variability and hydroclimate projections.



# 1 Introduction

The climate of the Common Era (CE), which spans the last two millennia, is considered the most thoroughly studied prein-
dustrial paleoclimatic period owing to the abundance of records from various paleoclimate archives (PAGES 2k Consortium,
2019) and climate observations. It provides rather stable conditions prior to the onset of industrialization with relatively con-
stant external forcing and close to present-day boundary conditions and thus also represents a well-studied benchmark for
climate models (e.g. Jungclaus et al., 2017). Over the last two millennia, the global climate presents itself as an interplay and
superposition of various major atmospheric and oceanic modes of variability. As such, different regions of South America
are influenced by the El Niño-Southern Oscillation (ENSO), the Pacific Decadal Oscillation (PDO), the Atlantic Multidecadal
Oscillation (AMO), the Southern Annular Mode (SAM) and variations in the position of the Intertropical Convergence Zone
(ITCZ) (Garreaud et al., 2009). The spatial and temporal variability of the South American Summer Monsoon (SASM) (Zhou
and Lau, 1998; Marengo et al., 2010) and its sub-component, the South Atlantic Convergence Zone (SACZ) (Carvalho et al.,
2004), create a wide range of climate zones across the continent. This makes the South American continent an intriguing
testbed for climatological research of the interplay of different phenomena before the onset of the current warming period.

The challenges posed by anthropogenic climate change are particularly pronounced in South America. The implications
for water resources are significant, given that South America comprises two of the world's most crucial river basins: the
Amazon Basin in the center-north and the Parana/La Plata Basin in the center-southeast. However, the broader effects of
anthropogenic climate change on the entire hydrological cycle and its variability, including changes in precipitation extremes
and the occurrence of droughts, remain widely unknown. Improving the understanding of the full range of climate variability
in South America is imperative, given the high vulnerability of human livelihoods in tropical and subtropical regions to the
impacts of climate change. This vulnerability is especially evident in extreme events like droughts and floods, due to both
geographic and socioeconomic factors (Pörtner et al., 2022), as seen for example in the current megadrought in central Chile
(Garreaud et al., 2019).

The climate of the past beyond the instrumental period is studied to enhance our understanding of hydroclimate and contex-
tualize recent changes. For instance, extreme events and their socioeconomic consequences have been recorded in historical
documents, such as a sequence of drought in northeastern Brazil (Aceituno et al., 2008; Utida et al., 2023), droughts in Bolivia
(Gioda and Prieto, 1999), and extreme floodings of the Parana River (Prieto, 2007).

Climate archives are increasingly used to reconstruct and interpret changes in climate beyond the instrumental era. These
reconstructions offer statistically robust estimates of the climate, particularly for the CE, the period of interest in our study. In
principle, existing global climate field reconstructions of the CE already provide estimates for both surface temperature and
hydroclimate variables, including for South America (e.g., Hakim et al., 2016; Franke et al., 2017; Steiger et al., 2018; Tardif
et al., 2019; Neukom et al., 2019). However, global CE reconstructions predominantly rely on proxy records from the mid-to-
high latitudes, particularly tree rings from the Northern Hemisphere. Climate proxy record density for the CE in tropical and
subtropical regions is much lower, in particular for terrestrial locations. Terrestrial proxy records are, however, crucial, when
it comes to reconstructing hydroclimatic variability. Although climate field reconstruction can make use of teleconnections to



alleviate data scarcity, the lack of local data limits the viability of global reconstruction in (sub-)tropical regions (Anchukaitis
and Smerdon, 2022).

For South America, climate field reconstructions are constrained by the scarcity of climate proxy records for all regions
except for the central and southern Andes, where tree rings serve as an abundant climate archive. Regional climate field
reconstructions have thus primarily focused on Southern South America (Neukom et al., 2010, 2011; Luterbacher et al., 2011;
Morales et al., 2020). However, in recent years speleothems have emerged as a promising climate archive with the potential to
alleviate data scarcity in tropical South America. Speleothems are geological cave formations created by accumulating layers of
calcium carbonates transported by seepage water. Among the many climate proxies archived in speleothems, the ratio between
heavy and light oxygen isotopes ($\delta^{18}$O) as saved in accumulating layers of calcium carbonate reflects the isotopic composition
of the precipitation above a cave and, thus, records hydroclimatic changes (Bradley, 2015). The $\delta^{18}$O signatures of precipitation
are sensitive to air temperature, precipitation amount changes, and the geographical location in terms of altitude, latitude, and
distance from the coast (Dansgaard, 1964).

Tropical South America is an archetypical region for speleothem research with a growing number of published records in
recent years. For instance, single records have been used to demonstrate changes in the intensity of the SASM on millennial
to centennial time scales in response to changes in orbital and solar forcing (Novello et al., 2016; Bernal et al., 2016). For
the hydroclimate of the last millennium, pronounced anomalies found in speleothem $\delta^{18}$O values have been associated to
the Medieval Climate Anomaly (MCA) and the Little Ice Age (LIA) (Novello et al., 2018; Apaéstegui et al., 2018; Azevedo
et al., 2019). Moreover, through the analysis of several South American speleothem records using dimensionality-reduction
techniques, Orrison et al. (2022) demonstrated that climate model simulations of the last millennium consistently underestimate
centennial climate changes over the South American continent, reinforcing findings of Rojas et al. (2016), who investigated
SASM variability in climate model simulations.

Nevertheless, these insights into South American Hydroclimate variability during the CE may be excluded from existing
climate field reconstructions due to the limited integration of speleothem records into these reconstructions. Their incorporation
proves difficult for two main reasons. First, reconstructions of the CE are usually attempted at seasonal or annual resolution and,
thus, only include proxy records of at least annual resolution. Speleothems are seldom dated annually with dating uncertainties
often on the scale of several years. Even so, karst processes above the caves work as smoothing filters of the isotopic variations
in precipitation. Thus, speleothem $\delta^{18}$O time series reflect the mixing of rainfall from different seasons and the transit time
through the epikarst to the water dripping point in the cave (for speleothems in Brazil see Moquet et al., 2016). Second, climate
field reconstructions usually require a calibration of the proxy records against instrumental data. This calibration is hampered
by the low temporal resolution of speleothem records and short data overlap with regional instrumental observations. Similar
restrictions also apply to many lake and marine sediments, precluding their use in current climate field reconstructions. In this
study, we aim to overcome these limitations to explore the information gain associated with including previously excluded
climate archives of annual to decadal resolution.

Here, we present the first climate field reconstruction of the hydroclimate of the South American continent for the entire CE,
employing speleothems besides more commonly-used climate archives such as tree rings, lake sediments, ice cores, corals,



sclerosponges, and marine sediments. We combine proxy records from a multitude of proxy record databases (Emile-Geay
et al., 2017; Comas-Bru et al., 2020; Konecky et al., 2020; Morales et al., 2020) and individual records provided by original
authors. This yields a total of 295 proxy records (See Figure 1). Our selection represents the at present spatially most complete
collection of publicly available proxy record data for the region.

As a climate field reconstruction technique, we choose Paleoclimate Data Assimilation (PaleoDA) (Bhend et al., 2012;
Steiger et al., 2014). Based on the principles of DA, which has been successfully applied in weather and ocean forecasting
for three decades (Evensen et al., 2022), PaleoDA fuses information from climate model simulations and climate observations
to provide a best state estimate. In contrast to other regression-based techniques (e.g Principal Component Regression as in
Luterbacher et al., 2002; Neukom et al., 2014), PaleoDA does not directly rely on gridded instrumental datasets. The climate
simulations provide time series long enough to also include proxy records of relatively low resolution, such as speleothems,
which cannot be calibrated to instrumental data. We use five state-of-the-art isotope-enabled climate simulations, which simu-
late the isotopic composition of precipitation and which were made publicly available recently (Bühler et al., 2022). This leads
to a twofold information gain: first, it enables the inclusion of speleothem records in the PaleoDA without the uncertainties as-
sociated with instrumental calibration, and second, it facilitates the comparison of multiple simulations of rainfall $\delta^{18}$O values
from different models, thereby mitigating biases stemming from individual proxies and models. In this study, we will focus
on surface temperature, precipitation amount, the drought index SPEI (Standardized Precipitation Evapotranspiration Index,
Vicente-Serrano et al. (2010); Beguería et al. (2014)), and the isotopic composition of precipitation. While in theory, PaleoDA
allows reconstructing climate fields for all simulated climate variables, the chosen variables are most closely related to the
climate signal recorded by the selected climate archives and, thus, hold the highest potential for reliable reconstructions. To
overcome the difficulties posed by proxy records of annual to decadal resolution such as speleothems, we adapt the PaleoDA
algorithm to a multi-timescale method enhancing the concept proposed by Steiger and Hakim (2016).

The structure of this study is as follows: We introduce the proxy record and climate model simulation data on which we base
our reconstruction. We then provide a complete description of the multi-timescale PaleoDA methodology, its assumptions and
key parameters. We present our obtained reconstruction (for both annual and austral summer means) and validate it by means
of comparison to instrumental data, other reconstructions, and non-assimilated proxy record data. As one of several climato-
logical applications of the reconstruction dataset, we study the main centennial hydroclimatological changes in South America
during the MCA and the LIA and compare our reconstruction to other studies. In addition, we investigate the reconstructed
intensity and variability of the SASM using a precipitation-based monsoon index, which has not been studied in climate field
reconstructions before.

## 2 Data

### 2.1 Paleoclimate Proxy Data

The continental climate field reconstruction provided in this study is based on a selection of different climate archive types.
While focusing on climate archives that store information about the hydroclimate, we also include other types of archives for



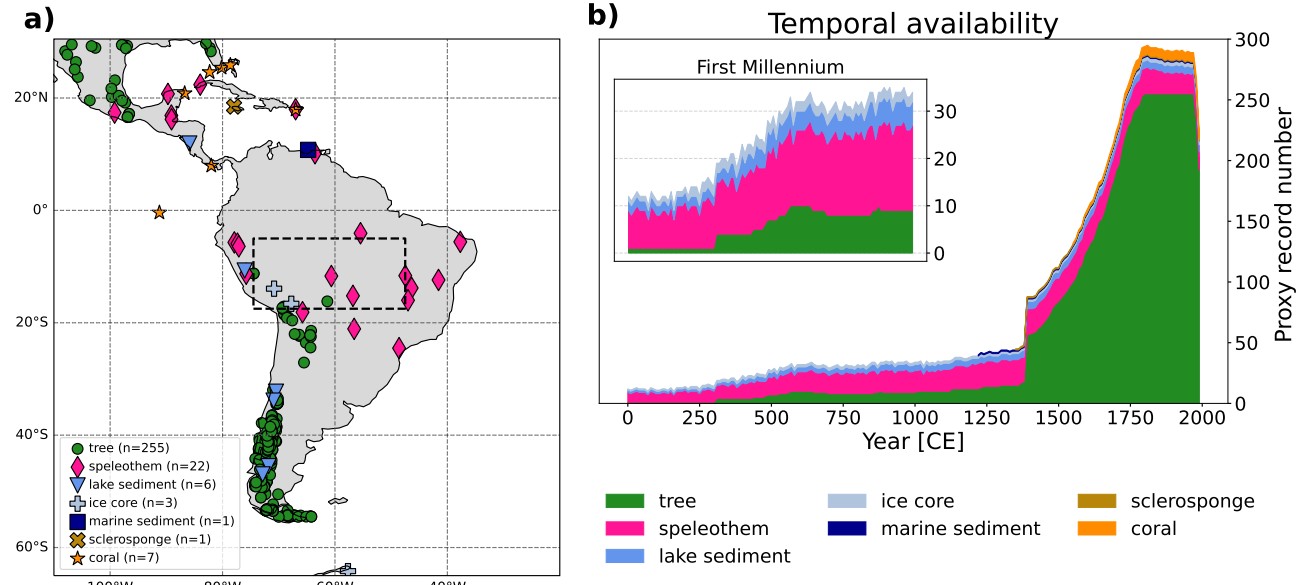

**Figure 1. Spatiotemporal availability of the employed proxy records**. a) Spatial distribution of all employed proxy records. Archive types are encoded by shape and color. The represented spatial domain (65°S-35°N, 20°W-110°W) is the spatial domain reconstructed in this study. In addition, the black box demarcates the core region of the South American Summer Monsoon (SASM) following the definition by Vuille et al. (2012), which we use to reconstruct the SASM index. b) Temporal availability of proxy archive types, with a zoom on the first millennium.

better covering the range of variables. The archives include trees, speleothems, ice cores, lake sediments, corals, sclerosponges and marine sediments. To ensure the best possible spatio-temporal coverage of the region, we have chosen and combined proxy records from three published paleoclimate proxy databases: PAGES2k (Emile-Geay et al., 2017) , Iso2k (Konecky et al., 2020),

125 the third version of the database from then Speleothem Isotopes Synthesis and AnaLysis group SISALv3 (Atsawawaranunt et al., 2018; Comas-Bru et al., 2020; Kaushal et al., 2023), and tree rings used for the South American Drought Atlas (Morales et al., 2020). We additionally included single proxy records which are currently not part of published proxy record databases (individual references in Appendix D). In contrast to the other proxy record databases, the SISALv3 speleothem database has not been employed in climate field reconstruction previously.

130 The SISALv3 database provides different sets of chronologies for the different speleothems. For this analysis, we use the authors' original chronology only. Further, age uncertainties, which are only provided for SISALv3, are not explicitly considered as these are mostly smaller than the timesteps in the multi-time scale PaleoDA algorithm (see Section 3.2). A description table for each climate archive type that we included, including metadata for the individual proxy records, the proxy variable, median resolution and methodological choices (Proxy System Model, assumed noise, seasonality, time scale) is provided in

135 the Appendix Tables D1 to D6.

We define four selection criteria for choosing proxy records from the respective databases:



1. The sites where the climate archives where obtained are located in the region 65°S-35°N, 20°W-110°W (See Figure 1). Proxy records from Central America are explicitly included as they can provide information for the northern part of South America, where few proxy records for the past two millennia exist.

2. The proxy variable is approximately linearly related to one of the climate variables provided by the climate model simulations (see Section 2.2 and 3.3). We follow the recommendations given by the authors in the original publications to decide if such a relationship holds. Note, that speleothem and ice core $\delta^{18}$O values can also be included in our proxy record collection because the climate model data also includes simulated isotopic-composition of rainfall.

3. The proxy records span at least 200 years and have at least one sample value in the period from 1750 to 1850 CE. This period is used as the reference period for computing anomalies of the proxy records relative to a shared time period. It has the advantage of including crucial records as speleothems, that lack data points during the more commonly chosen reference period of the 20th century. At the same time, this excludes many shorter records which would lead to high reconstruction skill during the instrumental era, but would not contribute to the reconstruction of centennial scale hydroclimate changes, and are, thus, acceptable to omit.

4. The resolution of the proxy records is better than 10 years (median), as we set 10 years as the largest reconstruction time scale in the multi-time scale PaleoData Assimilation approach (described in the Section 3.2). While decadal to multi-decadal and lower resolution is technically possible, the proxy records falling into that category are too coarsely resolved to contribute to the skill of the reconstruction of the past two millennia on multi decadal and centennial time scales, which is the focus of this study.

Our selection of proxy records for the South American reconstruction consists of 295 records covering southern and central America (Figure 1). Due to our selection criteria, the maximum number of proxy records is available in the reference period 1750 - 1850 CE and the least in the first century (13 records). Despite the decreasing number of available proxy records back in time during the CE, the spatial coverage remains fairly similar throughout time, as older proxy records are evenly distributed. This is particularly evident for tree rings, which are clustered in the Andes (see also AF1 for the spatiotemporal availability of proxy records for each reconstructed century). The temporal availability of proxy records shows a sharp decline in tree ring data in the year 1400 CE, coinciding with the start of the SADA tree ring database. Throughout the reconstruction, we remain attentive to potential artifacts resulting from this decline. Despite this concern, we anticipate only a small impact, given that the tree rings are situated in regions where there are already existing proxy records. Non-speleothem archives mostly cover the central and southern Andes and coastal regions, while speleothems significantly contribute to the coverage of tropical inner-continental regions over the entire two millennia. Regions lacking archive sites for proxy records can be found in the northern part of South America, namely Colombia, the Guianas and the north western states of Brazil. Additionally, the western part of the Southern Cone lacks proxy records. However, the South American Drought Atlas has demonstrated that tree ring records from the central and southern Andes can be skillfully used to reconstruct the hydroclimate of that region.



## 2.2 Isotope-enabled climate model simulations

We use five state-of-the-art isotope-enabled climate model simulations of the last millennium (Bühler et al. (2022) and Table 1). Additional information about boundary conditions and forcing parameters can be found in Bühler et al. (2022). Isotope-enabled climate models are essential for our study as they allow us to bypass the uncertain calibration of speleothem $\delta^{18}O$ records to temperature or precipitation. The isotopic composition of precipitation ($\delta^{18}O$ values) is an important hydrological marker that conveys additional information not captured by precipitation or temperature alone. From the simulations, we use

the variables surface temperature, total precipitation amount, and the isotopic precipitation. The drought index SPEI (Standardized Precipitation Evapotranspiration Index Vicente-Serrano et al. (2010)) is computed from simulated precipitation and temperature using the code based on the Thornthwaite's method for estimating potential evapotranspiration from the climate-indices package (Adams, 2017). The time scale is set to 12 months. We choose Thornthwaite's method over the more precise Penman-Monteith formula, because the available climate model simulation data does not include all variables required for the

latter. In parts of the analysis, we use a Multi-Model Ensemble (MME) by conducting separate reconstructions for each model prior and then computing the mean of the five reconstructions. In PaleoDA, MMEs are being used to provide more reliable reconstructions than provided by single model priors (Parsons et al., 2021; Annan et al., 2022; King et al., 2021, 2023). As we will use the climate model data only as anomalies, the purpose of the MME in this study is not to alleviate the effect of mean value biases in the models, but to provide a more diverse covariance structure, which constitutes the backbone of the PaleoDA

reconstruction algorithm (see Section 3.1). This covariance structure across models is particularly relevant in our study, as the covariance relationships between $\delta^{18}O$ and other climate variables as temperature and precipitation vary considerably in the different climate models. An MME requires regridding of all climate model simulations to the same grid. We choose to interpolate all climate model simulations to the highest resolution provided by one of our input models, isoGSM (1.875° x 1.875°). This allows for keeping the spatial information of the higher resolving models. Computing the correlation fields for

the regridded climate fields, we found that the regredding produces smoothed correlation fields without introducing artefacts (not shown). Note, that it is also conceivable to construct an MME by concatenating the model priors before applying the Ensemble Kalman Filter equations, but this method is not used here.

## 3 Methods

### 3.1 Data Assimilation with the Ensemble Kalman Filter as a Climate Field Reconstruction technique

Data assimilation (DA), which has been developed mainly for improving numerical weather and ocean fore- and hindcasts, combines two sets of information: a statistical prior state estimate that is provided by an ensemble of estimates from a numerical model, and information from observations (e.g. climate proxy records). The prior estimate is updated conditional on the observations (Evensen et al., 2022). The obtained posterior state is then propagated through time by the numerical model until new observations are available. PaleoDA often simplifies some aspects of DA compared to operational weather and ocean

forecasting systems, particularly by omitting the model ensemble restart step due to the limited predictability and the computa-



| Model | Resolution | Reference |
|-------|-----------|-----------|
| ECHAM5/MPI-OM | $3.75° \times 3.75°$ | Sjolte et al. (2020) <br> Werner et al. (2016) |
| GISS ModelE2-R | $2.5° \times 2°$ | Lewis and LeGrande (2015) <br> Colose et al. (2016a) <br> Colose et al. (2016b) |
| iCESM1 | $2.5° \times 1.875°$ | Brady et al. (2019) <br> Stevenson et al. (2019) |
| isoGSM | $1.875° \times 1.875°$ | Yoshimura et al. (2008) |
| iHadCM3 | $3.75° \times 2.5°$ | Bühler et al. (2021) <br> Tindall et al. (2009) |

**Table 1.** Spatial resolutions and references of the isotope-enabled last millennium simulations used in this study. The listed references also include the model descriptions. For a detailed description of boundary conditions and forcing parameters see Bühler et al. (2022).

tional cost of General Circulation Models (GCMs) on paleoclimatic timescales (Dirren and Hakim, 2005; Huntley and Hakim, 2010). AF2 illustrates the main steps of the PaleoDA algorithm.

Here, the updating operation is performed by the Ensemble Kalman Filter (EnKF) (Evensen, 1994), which assumes that the prior and the observations are sampled from the same true, but unknown Gaussian distribution. The errors with respect

to the true distribution are assumed to be unbiased and normally distributed. The EnKF further assumes that the observations are linearly related to the true distribution. Despite the normality assumptions, it has proven to be also a good estimator in non-linear cases where the assumptions do not strictly hold, which explains its wide use in real-word applications (Evensen et al., 2022).

The assimilation equation states

$$\mathbf{X} = \hat{\mathbf{X}} + \mathbf{K}(\mathbf{Y} - \mathcal{H}\hat{\mathbf{X}}), \tag{1}$$

where $\hat{\mathbf{X}}$ is the prior state (e.g. a surface temperature field over all simulated time steps, which creates an ensemble) and $\mathbf{X}$ the posterior state matrix. $\mathbf{Y}$ is the observation vector containing the proxy record values at a specific time step and $\mathcal{H}$ is the observation operator mapping the prior values to the observations (further explained in Section 3.3), thus creating $\mathcal{H}\hat{\mathbf{X}}$, the observation estimates for the simulated values. $\mathbf{K}$ is the Kalman gain matrix, a weighting matrix, which blends the prior state estimate and observations according to covariances in the prior ensemble and the proxy record uncertainty. It is computed as

$$\mathbf{K} = \mathrm{cov}(\mathbf{X}, \mathcal{H}(\mathbf{X}))(\mathrm{cov}(\mathcal{H}(\mathbf{X}), \mathcal{H}(\mathbf{X})) + \mathbf{R})^{-1}. \tag{2}$$

$\mathbf{R}$ is the observation error covariance matrix which contains the error associated with the proxy records (see Section 3.4). We assume, that the observation error of different proxy records is uncorrelated, hence $\mathbf{R}$ is a diagonal matrix. The EnKF assimilation equation can be effectively described as an interpolation process, where the model prior provides covariance

estimates between observation locations and all simulated locations to perform the interpolation. Equation 2 follows from



minimizing the posterior error covariance matrix, which is given by

$$\text{cov}(\mathbf{X}, \mathbf{X}) = (\mathbf{I} - \mathbf{K}\mathcal{H})\text{cov}(\hat{\mathbf{X}}, \hat{\mathbf{X}}). \tag{3}$$

The uncertainty of the reconstructed variables at each time step can be computed as the standard deviations of the diagonal entries of the posterior error covariance matrix (Equation 3). The posterior covariance matrix is by construction less than or
equal to the prior covariance. To solve the equations 1, 2 and 3 efficiently, we utilize the Ensemble Transform Kalman Filter (Bishop et al., 2001) as formulated in Vetra-Carvalho et al. (2018). The prior covariance matrix $\hat{\mathbf{X}}$ is used from already computed simulations instead of restarting the model ensemble and thus represents the climatological covariance. To compute it, we use an ensemble of 100 randomly selected simulation years. This approach has been named stationary offline PaleoDA (Okazaki et al., 2021) and assumes a stationary covariance between climate variables at the grid cells. Each year is recon-
structed separately, with the temporal pacing determined by the climate proxy records and the spatial information provided by the covariances from the climate model simulations. Note, that by using an MME (Section 2.2), we effectively use the mean Kalman gain from the five model priors. The offline PaleoDa concept is similar to computationally efficient Optimal/Statistical Interpolation methods (Evensen, 2003; Oke et al., 2005), although these still forward a single model simulation in time according to the results of the assimilation.

We reconstruct both annual (April-March, according to the vegetation cycle) and austral summer (DJF) means separately, due to the selected proxy records likely representing either annual or summer means (in particular speleothems from the SASM region).

In our reconstruction, the prior ensemble mean for each grid cell and climate variable is always enforced to be zero to limit model mean value biases. We consider this a valid approach, as the CE is a relatively stable climatic period. Not performing
such debiasing would introduce steplike, unphysical shifts in the reconstructed time series depending on the availability of specific proxy record with a strong difference towards the prior (as also noted in Franke et al. (2017) for example).

In the last decade, the EnKF has been introduced into the field of climate field reconstructions (Bhend et al., 2012; Steiger et al., 2014) and been used to reconstruct the climate of the last two millennia (e.g Hakim et al., 2016; Steiger et al., 2018; Tardif et al., 2019) and older periods such as the Last Glacial Maximum (Tierney et al., 2020; Annan et al., 2022), the last
deglaciation (Osman et al., 2021; Erb et al., 2022), and the Paleocene–Eocene Thermal Maximum (Tierney et al., 2022).

### 3.2 Multi-time scale Paleoclimate Data Assimilation

The PaleoDA algorithm described in the Section 3.1 is usually employed for the assimilation of proxy records at a single time scale, e.g. annual or seasonal for the CE. However, the speleothem records that are the backbone of our regional reconstruction have median temporal resolutions of 1 to 8 years and actually represent smoothed climate signals due to mixing effects of the
karst system on the cave drip water (Moquet et al., 2016). Therefore, we developed a new multi-time scale adaptation of the EnKF-PaleoDA algorithm, building on the concept proposed by Steiger and Hakim (2016), which allows us to use low- and high-resolution proxy records simultaneously in a computationally efficient manner.



The key idea is to add an additional time dimension for consecutive years in the prior state matrix **X**, enabling the use of multi-year means and covariances in the PaleoDA algorithm. The number of years added in the prior state matrix is determined

by the largest time scale imposed by the proxy records, which we determine as decadal for the employed speleothems. Instead of reconstructing the target time period (1-2000 CE) year by year, we divide it into decadal blocks, as we will use the speleothem values on a decadal time scale only. Additionally, we reconstruct annual and quinquennial time scales as sub-blocks of the decadal block. For each decadal block, the algorithm computes the assimilation equations for all proxy records that represent the largest time scale, using decadal means in the prior state matrix. The resulting decadal mean value is then exchanged in

the prior state matrix. The algorithm continues with the smaller time scales using the corresponding proxy records. Each proxy record is used on a single time scale. This procedure requires resampling the records to the target resolution of the desired timescale prior to the Data Assimilation. We first bin all the non-annual proxy records to the respective resolution using a simple equidistancing resampling routine, which consists of upsampling the time series values to annual resolution, filtering the time series with a low-pass filter and finally resampling to the targeted resolution (similar to the `MakeEquidistant`

function of the Paleospec R package (Laepple et al., 2023a)). We ensure the resampling does not add spurious datapoints by choosing target resolutions that are larger than the largest proxy record sampling interval and mask longer gaps in the proxy records. The multi-time scale algorithm involves more calculation steps than the single-time scale algorithm due to the repeated calculation of multi-year means and anomalies in the prior state matrix. As such, the algorithm also requires the repeated application of the observation operator $\mathcal{H}$ in the Kalman Filter equations. However, this is avoided by appending the

observation estimates to the prior state matrix, such that the algorithm also updates the observation estimates. A multi-time scale paleoclimate data assimilation approach is not only advantageous to allow for the assimilation of irregular proxy-records, it should also improve the reconstruction of multi-decadal to centennial climate variability according to pseudoproxy experiments (Steiger and Hakim, 2016; Choblet et al., 2023). Note, that this multi-time scale PaleoDA algorithm differs from the one used in the Holocene temperature reconstruction by Erb et al. (2022), where the multi-time scale prior ensemble is constructed as a

moving prior ensemble from transient climate simulations.

### 3.3 Proxy system models

In PaleoDA, Proxy System Models (PSM) (Evans et al., 2013; Dee et al., 2015) are employed for the observation operator $\mathcal{H}$ in equation 1 and 2. They have been developed to enhance model-data comparison by encapsulating the physical, geological, biological and biogeochemical processes into mathematical formulas to translate the external climatic conditions into a proxy

record signal. These processes are usually divided into three different stages, the sensor, archive and observation stage. In this study, we use PSMs in a one-stage manner, which is commonly chosen in PaleoDA studies targeting the CE climate (e.g. Hakim et al., 2016; Steiger et al., 2018; Tardif et al., 2019). Within this PaleoDA study, the PSMs translate the signal of simulated climate to proxy record units (the proxy record variance), accounting for seasonality. We employ three types of PSMs depending on the nature of the proxy records:





A) For records already calibrated to temperature, such as some lake and marine sediments and sclerosponges, the PSM takes the temperature of the model grid box closest to the proxy record location as the simulated temperature value. Seasonal means or annual means (April to March) are used depending on the indications in the original publication of the record. This way, seasonal biases in the reconstruction are constrained.

B) For proxy records that reflect changes in $\delta^{18}$O of precipitation, such as speleothems, ice cores, and some lake sediments, the PSM uses the simulated $\delta^{18}$O of precipitation values from the closest locations to the proxy records. Annual means of precipitation-weighted $\delta^{18}$O values are computed to account for varying precipitation intensity over the year. The values of the two ice core records given in deuterium are divided by 8 to represent the variability in $\delta^{18}$O according to the Global Meteoric Water Line (Craig, 1961). Although the $\delta^{18}$O values in these different archives is stored in different materials (e.g ice, carbonates, trees) and, thus, have different mean values, this is not relevant for our reconstruction, because we use proxy record anomalies respective to a reference period. We assume that temperature dependent fractionation effects are small compared to the variation in the $\delta^{18}$O of precipitation values.

C) For proxy records that can be calibrated to instrumental data, such as corals and tree rings, we employ a linear regression-based PSM (e.g. Hakim et al., 2016; Steiger et al., 2018; Tardif et al., 2019). Linear regression equations are estimated between the proxy time series and instrumental time series over a calibration period. The regression parameters are then applied to the model data in the data assimilation. Different predictor variables, such as surface temperature, precipitation, or the SPEI drought index, are used based on the lowest p-value for each proxy record. For coral records, we only use surface temperature as a predictor variable. As instrumental calibration datasets, we use the Berkeley Earth dataset for surface temperature (Rohde and Hausfather, 2020), precipitation from CRUTS 4 (Harris et al., 2020b) and an SPEI drought index computed from the temperature and precipitation in CRUTS 4. The temperature calibration is performed over the period 1920 - 2000 CE. For precipitation and SPEI, we use the period 1950 - 2000 CE due to limited local station data in South America before 1950 CE (Garreaud et al., 2009). These spatially highly resolved instrumental datasets have been regridded to the spatial resolution used in the climate field reconstruction to account for the lower spatial resolution of the climate model simulation data. We compute these regressions both for the annual and seasonal instrumental means and choose the best predictor variable separately (as in Steiger et al., 2018). For the annual reconstruction, 110 tree-ring records were predicted with temperature, 94 with SPEI and 51 with precipitation. For the austral summer reconstruction, 102 tree-ring records were predicted with temperature, 92 with SPEI and 61 with precipitation.

## 3.4 The proxy record error

The proxy record error in Data Assimilation represents the non-climatic noise recorded by proxy records and is challenging to quantify. Measurement noise is considered negligible. For the tree and coral archives, where calibration could be performed with instrumental data, the proxy error follows directly from the linear regression in the PSM (See Section 3.3). The mean of the squared linear regression residuals is the proxy error variance. For all other records in the non-instrumental era, we express the proxy record error in terms of the signal-to-noise ratio (SNR), assuming Gaussian and timescale-independent noise



(Smerdon, 2012). The SNR can then be converted into the entries of the observation error matrix $\mathbf{R}$ in Equation 2 by taking into account the variance of the proxy records, $\mathrm{var}(\mathbf{Y})$, (see Appendix C for the derivation):

$$\mathbf{R} = \mathrm{var}(\mathbf{Y})/(1 + \mathrm{SNR}^2) \qquad (4)$$

$\mathbf{R}$ is computed for each proxy record that can not be calibrated using its resampled time series (see Section 3.2). Given that some proxy records, particularly speleothems, exhibit significant shifts in local climate or environment, we observed that the variance and thus $\mathbf{R}$ would be larger compared to those not indicating such local changes. Consequently, the former records exerted less influence during the reconstruction. To address this issue, we used the mean of the variance in a running 200-year

window for computing $\mathrm{var}(\mathbf{Y})$ to reduce the influence of shifts on the variance and to ensure comparable noise levels for all proxy records. We thus assume, that for a 200 year window, the statistical assumptions of Gaussian and timescale-independent noise leading to equation 4 hold. We assume an SNR of 0.5 for all proxy records which we have not calibrated to instrumental variables, following exploratory studies by Wang et al. (2014) and Orrison et al. (2022). We emphasize that the proxy record error should not be considered separately from the variance in the prior ensemble matrix, which represents the model error (see

Equation 1). The relationship between proxy error and prior variance is crucial in PaleoDA. While in regular Data Assimilation via the optimal interpolation method, the static prior variance is usually considered to be too large and thus reduced by a factor (e.g. Oke et al., 2005), climate simulations have been allegedly underestimating climate variability, for instance in surface temperature (Laepple and Huybers, 2014; Laepple et al., 2023b) or isotopic variability in precipitation (Bühler et al., 2022), which could be used as an argument in favor of inflating the variances in the prior. Instead of adjusting the variance in the prior,

we also performed alternative reconstructions in which the proxy error variance for non-calibratable records was set equal to the variance in the prior observation estimates. This approach gives equal weight to the proxy observations and the model prior.

### 3.5 Further reconstruction refinements

We use a Monte Carlo technique of repeating the reconstructions 50 times with different ensembles of 100 randomly selected simulation years and using 80% of all proxy records in each repetition similar to Hakim et al. (2016) and Tardif et al. (2019).

Doing so improves the representation of the reconstruction uncertainty and attenuates the effect of outliers in the proxy record selection. The withheld proxy records will be used for internal validation of the reconstruction. Covariance localization, which is used to suppress spurious long range covariances is not employed in this study, because the prior ensemble size is considered large. Furthermore, PaleoDA studies that use covariance localisation usually do so with very large decorrelation lengths larger than 12000 km (e.g. Tardif et al., 2019; Tierney et al., 2020; Osman et al., 2021), exceeding the targeted area in our

reconstruction.

### 4 Validation

We validate our reconstruction using gridded instrumental data sets from the 20th century, independent reconstructions and proxy records withheld from the reconstruction. The focus here is put on the internal validation using withheld proxy records





as performed in previous PaleoDA studies (e.g. Tardif et al., 2019; King et al., 2021; Tierney et al., 2020; Osman et al., 2021).
The reason for doing so is that although instrumental validations are commonly used in PaleoDA and most easy to interpret, we do not consider them representative in our case. The instrumental temperature and precipitation time series are too short for validating a decadal-scale reconstruction. The 20th century validation mainly reflects the reconstruction capability of tree rings, which are the most abundant climate archive in the instrumental period but only provide limited spatial and temporal coverage during the entire CE. An extensive validation using gridded instrumental temperature and precipitation data for evaluating the
reconstruction regionally and for the SASM precipitation amount is presented in Appendix B1 and B2. A validation of the reconstructed drought index for the Southern Cone, for which independent reconstructions exist is presented in Appendix B3.

Withholding proxy records in each repetition of the repeated Monte Carlo reconstructions allows us to perform an internal validation of the reconstruction (see Section 3.5). The proxy record estimates from the simulation data (e.g. the simulated
isotopic composition of precipitation of the grid cell in which a cave is located) are still part of the prior state matrix, also when a proxy is not used as input data and, therefore, updated in the PaleoDA algorithm. We calculate the correlation and Coefficient of Efficiency (COE) (Nash and Sutcliffe, 1970) of the withheld proxy record time series to their reconstructed counterparts. While the correlation rewards a correct timing of the reconstructed values, the COE is also sensitive to bias and errors in the amplitude by taking into account the variance of the true signal. The COE can take values in the range [-∞,1], with positive
values indicating skill. The correlation and COE are computed as the mean correlation for all Monte Carlo repetitions in which a proxy record is not used, by averaging the reconstructed time series to the respective temporal resolution of each proxy record. As we performed 50 Monte Carlo reconstructions with 20% of withheld proxy records, there are on average 10 reconstructions for each proxy record for which it has not been used as input data. We also compute these scores for the prior model simulations and compare the obtained values to those of our reconstruction. This allows us to assess if the reconstruction is in accordance
with independent proxy record data and indirectly identify regions in which proxy data share a common climate signal and are similarly reconstructed. The results are displayed in the form of histograms and on maps in Figure 2.
We obtain predominantly positive correlations (82.03%), with a median value of 0.21. Most correlations are positive and significant (60.34%). As the correlations of the model simulations to the proxy records are negligible, the reconstruction does lead to a clear improvement in correlation with a median increase of 0.22 and 81.02% of the records with an improvement.
However, a few records stand out with negative correlations in the reconstruction, particularly in geographically isolated areas, while higher correlations are generally obtained in regions with numerous proxy records. For the COE, the obtained median value is -0.01. Yet, 43.05% of the values result in a positive COE. In comparison to the model priors, there is a notable median increase of 0.34, with 89.83% of records demonstrating an improved COE value. The COE score results underscore the high dissimilarity of proxy records in northern and eastern Brazil, possibly linked to the climate dipole between northeastern
Brazil and the core SASM region (Novello et al., 2018; Campos et al., 2019; Wong et al., 2021), where speleothem records exhibit opposing $\delta^{18}$O trends. This spatial homogeneity in the reconstruction may potentially mask this crucial feature of South American climate. Such homogeneity is expected due to the coarse model grid resolution, which also dictates the reconstructed resolution.



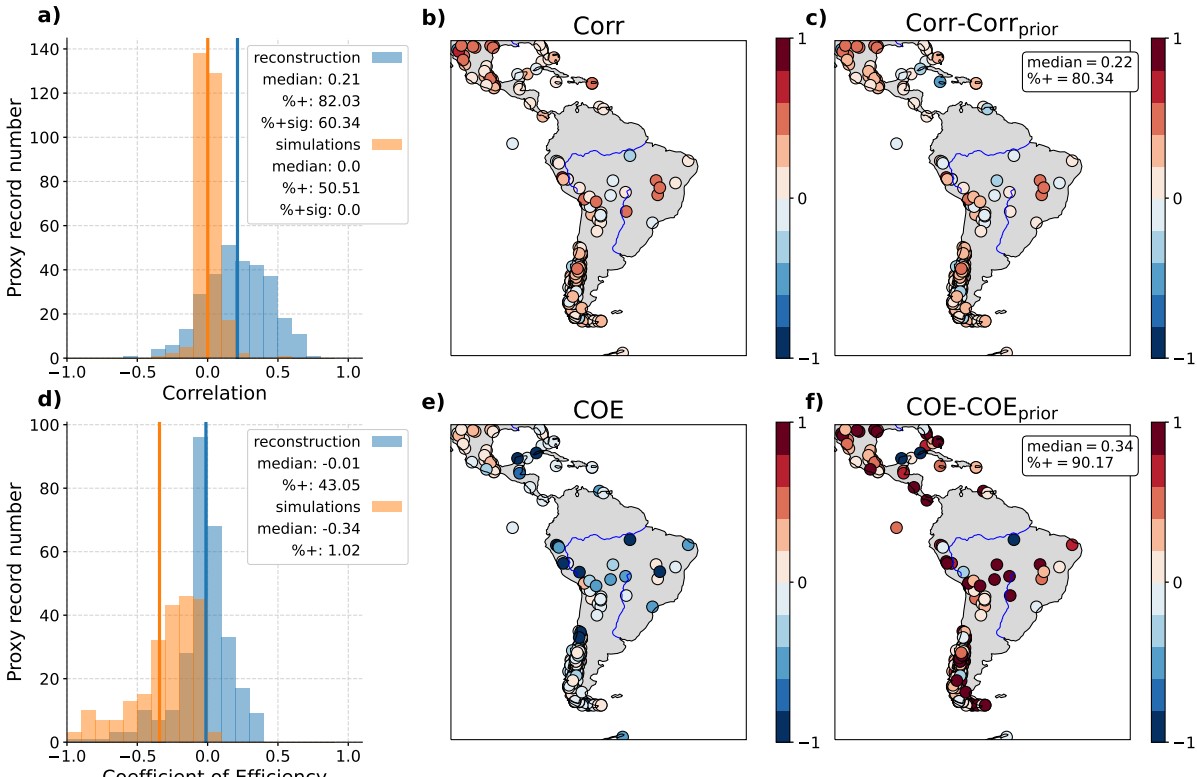

**Figure 2. Internal validation**. We compute the correlation (a,b,c) and Coefficient of Efficiency (COE) (d,e,f) of reconstructed proxy record time series to true proxy record timeseries when the proxy records are not used as input data in the reconstruction (withheld proxy records). Panel a) and d) show the histogram of the skill scores. The box in the upper right corner displays the median value, the percentage of positive values (%+) and the percentage of positive and significant correlations (%+sig, p-value $< 0.05$) for the distributions. Panels b) and e) display the skill scores for each location. We additionally computed the skill scores for the model simulations (mean of the five models for each proxy record), which are also included in the histograms and used for a comparison between reconstruction skill scores and the model simulations (c,f). Note, that the COE can also take values smaller than -1. In effect, 24% of the COE values for the simulations fall outside of the histogram. While all colorbars have been limited to the range (-1,1) for clarity, values in e) and f) can be more negative than -1, and the difference between reconstruction and simulations in f) also larger than 1.

While the skill values appear low, they are comparable to those obtained in the global multi-proxy reconstruction by Hakim
et al. (2016); Tardif et al. (2019); King et al. (2021). Furthermore, the comparison to skill scores of the raw simulation data underlines that the assimilated product represents the climate signal of the proxy records better.



# 5 Results

## 5.1 Centennial climate changes

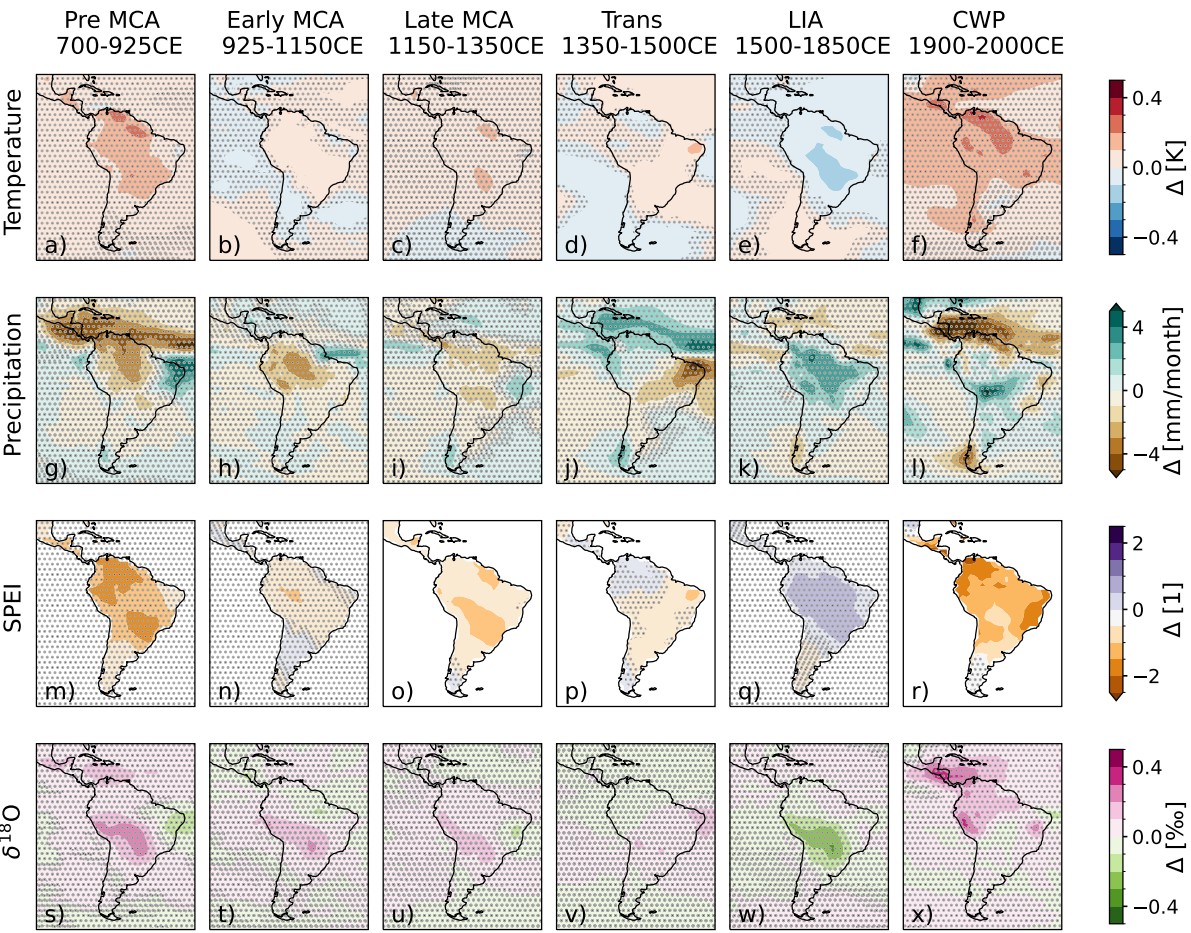

**Figure 3. Reconstructed mean anomaly fields** for temperature (a-f), precipitation (g-l), SPEI (m-r) and $\delta^{18}$O (s-x) during five periods with respect to the Last Millennium mean (LM, 850 - 1850 CE). The studied periods are the years preceding the Medieval Climate Anomaly (Pre MCA, 700-925 CE, first column), the early MCA (925-1150 CE, second column), the late MCA (1150-1350 CE, third column), the transition period (Trans, 1350 - 1500 CE, fourth column), the LIA (1500 - 1850 CE, fifth column) and the current warm period (CWP) (1900-2000 CE, sixth column). The SPEI values have been standardized using the variance of the period 850 - 1850 CE. Stippling indicates grid cells where the difference to the Last Millennium values is not significant according to a Welch's t-test ($\alpha > 0.01$).

Having examined the potential and limitations of our algorithm we now examine the climatic features displayed by the
reconstruction. As a first application, we analyze the newly reconstructed climate anomalies during the MCA and the LIA. The definition of these anomalous climate periods is equivocal, as they have not occurred synchronously globally (Neukom





et al., 2019). Therefore, we decide to examine finer intervals, namely the period preceding the MCA (Pre MCA, 700 - 925 CE), the Early and Late MCA (925 - 1150 CE, 1150 - 1350 CE) as in Azevedo et al. (2019), a transition period (Trans, 1350 - 1500 CE), the LIA (1500 - 1850 CE), and also the current warm period (CWP) (1900 - 2000 CE) to put the anomalies into
context. Figure 3 shows the anomalies for the annual reconstruction with respect to the mean of the last millennium (LM, 850 - 1850 CE). The same figure for austral summer (DJF), which overlaps with the monsoon period is displayed in AF3 and for the alternative proxy error definition in AF4. The centennial evolution of the mean states for the entire CE is shown in AF5 to AF8. The reconstructed patterns are mostly homogeneous over the continent and show a trend towards colder and wetter conditions during the LIA, especially for the central and northern part of the continent. The Southern Cone, however, experienced warmer
and drier conditions during the LIA. Warmer and drier conditions are predominant during the transition period, in particular preceding the MCA, except for the Southern Cone and the Nordeste (North Eastern Brazil). In the reconstruction using the equal variance proxy error definition (AF4), trends with similar pacing are observed, yet exhibiting greater strength. Remarkably, the important changes of the hydroclimate state before the CWP are mostly conveyed by the speleothem proxy record information, as reconstructions relying only on speleothem data and without speleothem data reveals (AF9 and AF10).

We observe in-phase trends for all reconstructed climate variables and studied periods, generally indicating a simultaneous occurrence of warmer with drier conditions and colder with wetter periods, except during the CWP. Compared to the last millennium mean, the reconstructed CWP anomalies show a more spatially diverse precipitation anomaly field with increased precipitation for the center of the continent. Less precipitation is reconstructed for coastal locations in the north, east, and southern margins of the continent. Temperature and SPEI, in contrast, show warmer and drier conditions for the entire continent,
except parts of the La Plata basin and the Southern Cone. The austral summer reconstruction (AF3), however, shows the largest positive temperature anomaly for these regions. For both annual and austral summer reconstructions, the 20th century is warmer and drier than all preceding phases. In terms of spatial homogeneity, the temperature and SPEI anomalies are more extensive compared to the precipitation reconstruction, which has more diverse spatial features. The reconstructed $\delta^{18}$O of precipitation values for the studied periods change most in the center of the continent, with a trend towards most depleted precipitation
during the LIA. The center of the largest changes in the isotopic composition of precipitation is located further to the south than the region of the largest precipitation changes. The $\delta^{18}$O values of precipitation display a dipole pattern over the Nordeste and central South America, except during the more homogeneous MCA-LIA transition period and CWP, where a pronounced $\delta^{18}$O enrichment is seen for the northern part of the continent.

## 5.2 South American Summer Monsoon variability

To assess changes in the SASM strength throughout the past two millennia, we analyze the mean precipitation anomaly in the core monsoon region as a simplified monsoon index. We use the definition by Vuille et al. (2012), who proposed computing the mean precipitation in the core monsoon region (5° S–17.5° S/72.5° W–47.5° W, see black rectangle in Figure 1) as an indicator of monsoon strength. Additionally, we further investigate $\delta^{18}$O, temperature and SPEI changes in the core monsoon region. Figure 4 shows the anomalies from the annual reconstructions in comparison to the reconstructions LMRv2.1 (Tardif
et al., 2019) and PHYDA (Steiger et al., 2018). We also performed the reconstruction with subsets of the complete proxy record





**Figure 4. Mean anomalies of precipitation, $\delta^{18}$O , SPEI, and temperature in the core monsoon region** ($5° - 17.5°$ S / $72.5° - 47.5°$ W). Comparison of reconstructed mean annual precipitation (panel a), $\delta^{18}$O (panel b), SPEI (panel c) and temperature (panel d) anomalies in the core monsoon region from reconstructions using all proxy records, excluding the speleothems, using only speleothems and only trees and the reconstruction products PHYDA (Steiger et al., 2018) and LMRv2.1 (Tardif et al., 2019). All time series have been smoothed with a 50-year-lowpass-filter. The anomalies are displayed with respect to the 850 - 1850 CE mean. Note the inverted y-axis for $\delta^{18}$O in order to match the precipitation trend. PHYDA and LMRv2.1 do not include all variables studied here, and are thus partially missing in panel a), b) and c). The SPEI values have been standardized using the variance of the 850 - 1850 CE period. For means of comparability between different reconstructions, we have chosen to display the annual mean reconstruction.



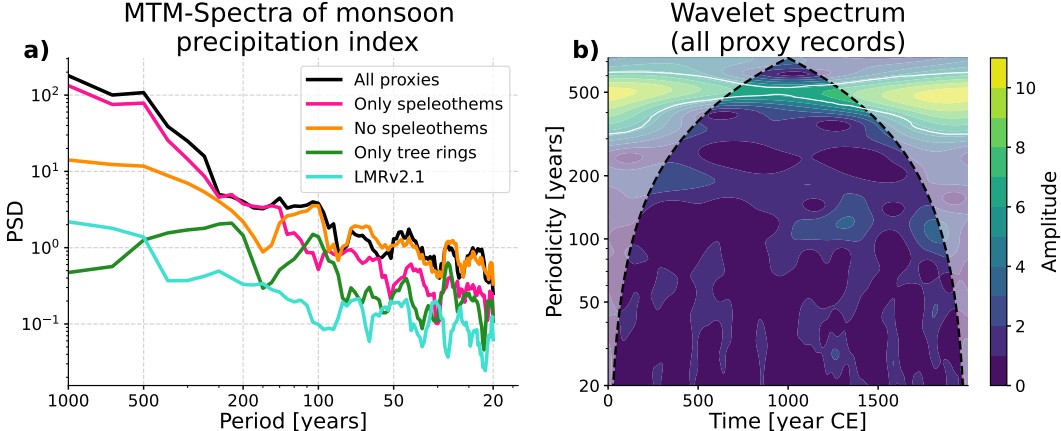

**Figure 5. Spectra of reconstructed monsoon precipitation index**. On the left (a), the Multitaper method (MTM) - power spectral densities (PSD) of the annual reconstructions involving different subsets of the proxy record database. The time series have not been standardized and detrended, but resampled to 10 year averages to achieve comparability between reconstructions involving different time scales. On the right (b), the continuous wavelet spectrum for the reconstructed time series involving all proxy records is shown. The dotted black line indicates the cone of influence and the white lines indicate the 95% significance level for an AR1 process. All spectra have been computed using the Pyleoclim package in Python (Khider et al., 2022).

database, only relying on speleothems or tree data, or excluding the speleothems from the complete proxy record database to investigate the effect of using different climate archives.

Overall, the reconstructed monsoon index in Figure 4 shows a colder and wetter LIA, particularly after 1500 CE with anomalies of up to 4 mm/month in the lowpass-filtered curve. Least precipitation is reconstructed for the period from 750 - 430   1100 CE, while the SPEI values reach a local minimum earlier in the period 600 - 900 CE, and thus seem to more strongly follow the temperature curve, which reaches a local maximum during the same period. Reconstructed temperatures are highest and SPEI - SASM indices lowest during the 20th century. This hints at the effects of anthropogenic warming being evident in the core monsoon region on a centennial time scale. Figure 4 shows that the tree-only reconstruction does not reflect the pronounced centennial hydroclimate variability, although the tree ring data represents the dominant climate archive in our 435   proxy record database in terms of numbers. Comparing the reconstruction that exclusively uses speleothem records to the one that excludes speleothems reveals reconstructed hydroclimate changes in the core monsoon region to be largely driven by the speleothem signal. However, wetter LIA conditions are also captured by other archives. The same figure for the austral summer reconstruction and the *all proxies* estimates of the single model priors can be found in AF11 and AF12. We note that the presented results draw on mean values of the MME reconstruction. As visible in AF12, the magnitude of reconstructed 440   changes can vary considerably between the single model reconstructions, in particular for precipitation.

Figure 4a additionally displays the LMRv2.1 precipitation reconstruction, which shows lower centennial-scale variations, including no significant changes during MCA, LIA or CWP. Precipitation is not included in the PHYDA reconstruction;



however, the comparison of reconstructed SPEI also presents less hydroclimate variability during the LIA and the pre and early MCA phase. The temperature reconstruction in Figure 4c shows the largest range of values for the PHYDA reconstruction,
followed by the reconstructions of this study, and lastly the LMR reconstruction, which shows the least temperature variability. PHYDA shows a constant temperature decline during the CE with a steep reversal during the 20th century, resulting in the expected hockey stick-like curve, which is also found in our reconstruction, except before 400 CE. The period before 400 CE is peculiar for our reconstruction in all four variables. It shows very wet and cold conditions for the first two centuries of the CE and a subsequent transition to more neutral conditions. The reconstructed extremes even exceed the LIA in magnitude.

For the austral summer SASM index reconstruction (AF12), the range of values for the precipitation anomalies is almost twice as large as for the annual reconstruction, although the overall trends are very similar. Comparing the all proxy record reconstruction to the prior model simulation shows that while the model simulations do exhibit more important fluctuations, they do not exhibit clear trends, for instance of a cooler and wetter hydroclimate during the LIA (AF13).

To further study and quantify climate variability in the core monsoon region we calculated the power spectral distributions
of the precipitation anomaly curves (Fig. 5a). In addition, the continuous wavelet spectrum of the SASM precipitation reconstruction involving all proxy records was computed to investigate the power variation in time over the common era. The same spectral analysis for the reconstructed $\delta^{18}$O signal in the core monsoon region can be found in AF14. The precipitation spectra have typical *red noise* characteristics, with the power decreasing exponentially towards higher frequencies, except for the reconstruction only based on tree data, which has a flat power spectrum. The reconstructions including the speleothems show the
highest redness with a decline in power over three orders of magnitudes, as also indicated by the pronounced multi-centennial variability in Figure 4. Without the speleothems, the power distribution extends over two orders of magnitude. The power spectrum of the SASM precipitation anomaly in the LMRv2.1 reconstruction, remains flat and similar to the spectrum of the tree-ring only reconstruction. The wavelet spectrum of the all proxy precipitation reconstruction also exhibits multi-centennial variability of the SASM (Fig. 5b), although it is not above the signifance threshold. The spectra for the $\delta^{18}$O-SASM index,
which might be represented more directly by the available proxy records show similar *red noise* characteristics (AF14). For further comparison, we compared the spectrum of the SASM precipitation index to the spectra of the index in the prior model simulations. To do so, we took into account that the model simulations span a shorter time period than the entire reconstruction period and that they have a higher variance than the reconstruction. The spectra of all five model simulations have a flat shape and thus a smaller variability scaling than the reconstruction (AF15).

# 6 Discussion

## 6.1 Reconstructed hydroclimate changes

Our climate field reconstruction of the South American Hydroclimate during the CE represents a comprehensive synthesis of an encompassing diverse collection of proxy record data and isotope-enabled climate simulations. The reconstructed centennial climate changes in tropical South America, transitioning from a drier MCA to a wetter LIA, align well with individual proxy
record assessments from the region (Bird et al., 2011a; Vuille et al., 2012; Deininger et al., 2019) and proxy record syntheses



(Campos et al., 2019; Orrison et al., 2022). This consistency is expected, given that the temporal pacing of the reconstruction relies on the information provided by the proxy records within the offline PaleoDA algorithm.

The PaleoDA climate field reconstruction offers the advantage of extrapolating proxy record information to more climate variables, such as temperature from speleothem $\delta^{18}$O for instance, which in this region is mainly only interpreted in terms

of precipitation amount changes. The magnitude of reconstructed precipitation amount and SPEI surpass the estimates from the PHYDA and LMRv2.1 reconstructions, while the changes in temperature show similar magnitudes. It is worth noting that the highest temperature anomalies are reconstructed for the period preceding the MCA reference period 950 - 1250 CE (Fig. 5 and AF5). This pattern is also observed in the PHYDA and LMR reconstructions (AF16 and AF17), which rely on fewer local proxy record data points from South America for that period. In line with Neukom et al. (2019), South America does

not show coherent warming during the MCA compared to other regions globally, which contrasts with the qualitative study by Lüning et al. (2019). Although our reconstruction includes anomalous conditions, their timing might differ from the Northern Hemisphere anomalies. Thus, this reference time period should be used with caution in global comparisons.

Another similarity between our reconstruction and PHYDA and LMRv2.1 is the presence of anomalies of opposite signs in the Southern Cone region compared to the Northern and Central part of the continent. The influence of the Southern Annular

Mode in this region may, for instance, cause this distinct opposite climate system. Our reconstruction also reveals a spatial feature over northeastern Brazil, where opposite anomalies in precipitation, drought index and $\delta^{18}$O are observed compared to the rest of tropical South America. This dipole pattern, previously identified in regional speleothem studies, has been linked to meridional changes in the Hadley cells (Cruz et al., 2009; Novello et al., 2012, 2018) and interhemispheric temperature gradients (Campos et al., 2022). The presence of this dipole pattern in our reconstruction was not necessarily expected, considering

the limited number of speleothems from the Nordeste and the relatively coarse spatial resolution of the climate models. In addition, looking into the spatial correlations of the mean $\delta^{18}$O of precipitation values for that region in the model simulations does not reveal a clear dipole in all simulations (AF18) and thus reveals the benefit of combining proxy and model data with PaleoDA.

A consistent pattern in our hydroclimate reconstruction are the synchronous trends for all variables during most of the CE,

except for the 20th century. However, it is essential to acknowledge that the reconstruction technique may also influence this in-phase relationship. Offline PaleoDA with the EnKF, being inherently linear, employs the same static covariance patterns from the simulations at each time step. The in-phase relationship represents the default state of the reconstruction. The increased strength of the SASM during the LIA has been explained by cooler temperatures on the Northern Hemisphere, resulting in a southward shift of the ITCZ (Deininger et al., 2019; Vuille et al., 2012). Integrating our employed proxy records into a global

climate field reconstruction, and possibly including more atmospheric variables, could provide a more in-depth understanding and allow for testing this hypothesis.

An unexpected finding in our reconstruction is the evidence of wetter and colder conditions prior to 400 CE. Primarily, the speleothem data support these changes. Upon closer examination of the individual anomalies of the available records in AF19, only three records exhibit particularly negative $\delta^{18}$O anomalies for that period. Also, during that time, proxy records

from the margin and outside the core monsoon region are the only available sources. Still, the spatial correlations in the model




simulations for the precipitation in the core SASM region prove to be extensive, such that the signal from proxy records outside of the region can influence the reconstructed SASM precipitation (AF20). The increased SASM strength stays consistent even after excluding possibly diverging single records that were suspected to cause the anomalies in the reconstruction.

## 6.2 Comparison to reconstructions from single proxy records

Putting attention to particularly anomalous proxy records reveals further insights into the nature of our reconstruction. For instance, the record with the strongest negative anomaly from Paraiso Cave (SISAL-ID 424) in the eastern Amazon basin (Ward et al., 2019; Azevedo et al., 2019) exhibits a distinctive increase in $\delta^{18}$O values after 900 CE, resulting in negative $\delta^{18}$O anomalies during the early Common Era compared to the reference period (1750-1850 CE). This increase may be caused by non-climatic influences, which potentially introduces artifacts in our anomaly reconstruction. The record also stands out in

the internal validation procedure (Figure 2). Nonetheless, this record has been cross-validated with archaeological evidence from the region, which also suggests a shift from wetter conditions in the first millennium to drier conditions in the second millennium (de Souza et al., 2019). In contrast to other speleothem locations in tropical South America, the Paraiso Cave record is additionally influenced by the rainfall of the two distinct systems, SASM and ITCZ. Our PaleoDA approach may not be able to resolve this complex local climate. Different to the drier conditions recorded in Paraiso Cave (Ward et al., 2019),

the reconstruction does not show drier conditions during the LIA for the eastern Amazon basin, but wetter conditions as for entire tropical South America. While this record has an important influence in the first centuries of the reconstruction, it is outweighed by the multitude of other records during the LIA.

The contrast between reconstructed wetter and colder climate anomaly fields during the LIA to the information provided by single records highlights the spatial smoothing caused by the PaleoDA method, which we consider the central limitation

of our study. The EnKF employs spatial covariances between grid cells and variables in the climate model simulations, which tend to be spatially extensive when computed over the last millennium. As a result, the reconstruction for the eastern Amazon basin displays wetter conditions, because this is the condition recorded by the majority of the proxy records. While the PaleoDA methodology aims to compute the best climate field estimate from all observations, this example demonstrates that the reconstruction may not always align well with individual observations. This spatial smoothing effect due to the model covari-

ances has been noted in PaleoDA literature before, e.g., Sanchez et al. (2021), who focused on ENSO reconstructed from coral records during the 19th century, or Erb et al. (2022) who reconstructed the entire Holocene (see the outstanding proxy record anomalies in their Figure 10). In statistical terms, PaleoDA with the EnKF may cause a loss of spatial degrees of freedom in the reconstruction, which can be expressed in terms of the explained variance of the leading modes obtained via a Principal Component Analysis (Bretherton et al., 1999). We are not aware of studies investigating climate field reconstructions under the

aspect of spatial degrees of freedom, and doing so would be out of scope for this study. However, we note that when investigating spatial temperature correlations of CE reconstructions, Bakker et al. (2022) found larger inter-continental correlations for a PaleoDA reconstruction product compared to other methods (see their Figure 5). Although our reconstruction product is provided at a spatial resolution of 1.875° x 1.875°, it is more likely to reliably represent spatially extensive regions rather than single grid cells. This aspect is further emphasized by the fact that the climate model simulation data was up-sampled to the



resolution of the model with the highest spatial resolution.

Despite using five different isotope-enabled climate model simulations to mitigate the effect of single model biases, an ensemble of five models still represents a small ensemble of opportunity. Further, water isotopes have not been included in climate model intercomparison projects, and the number of publicly available simulations is limited to the five models we have employed. The reconstruction would importantly benefit from more isotope-enabled simulations with a variety of models and
more up-to-date forcing, such as volcanic forcing (e.g Sigl et al., 2015). Future PaleoDA reconstructions will also profit from explicitly studying the differences in spatial covariances simulated by the models to potentially discard overly or undersensitive models (e.g., in the covariance relationship of $\delta^{18}O$ values and precipitation changes). Multi-model reconstructions should further account for similarities between models in the form of weighted ensembles (Eyring et al., 2019). Alternatively, perturbed physics ensembles of isotope-enabled simulations for a single model would allow for larger ensembles and a transient offline
PaleoDA approach as used by Franke et al. (2017) and Valler et al. (2020). The high computational cost of including water isotopes in the simulations currently restrains this type of experiment.

PaleoDA reconstructions going back deeper in time have also employed prior ensembles that are moving with time to account for changing boundary conditions and, thus, the non-stationarity of simulated covariances (Osman et al., 2021; Erb et al., 2022). However, the time scales on which covariance patterns of specific hydroclimate variables in the climate system
change have not been studied formally yet.

### 6.3   Multi-decadal hydroclimate variability in the reconstruction

We also studied the reconstructed hydroclimate variability to assess the impact of different proxy archives, particularly speleothems. The power spectral distributions of the reconstructed SASM precipitation strength underscore that the use of different climate archives yields distinct hydroclimate variability patterns. Remarkably, the inclusion of speleothems lead to substantial variabil-
ity increase on multi-decadal to centennial time scales, contributing to a potentially more realistic reconstruction of the South American hydroclimate.

The signfcant multi-centennial variability of the SASM (Fig. 5) is comparable, albeit less pronounced and persistent, to findings in individual speleothem records (Novello et al., 2012; Apaéstegui et al., 2014; Novello et al., 2016; Deininger et al., 2019), where this variability was attributed to solar cycles of 83 and 208 years. However, it is essential to consider that the
reconstructed 2000-year time series is probably too short to definitively establish causal relationships for multi-centennial variability. Previous research found only small attributions of solar forcing over the last millennium (Schurer et al., 2013). Future PaleoDA reconstructions using speleothem data may focus on longer reconstruction periods to more thoroughly investigate links to solar forcing.

The resemblance between the reconstructed hydroclimate variability in our study and the estimates from single proxy record
studies is expected, given that we use data from these studies as input. However, PaleoDA enables us to underscore the connection between the isotopic composition of precipitation and precipitation amount. Moreover, spectral analyses derived from proxy record compilations generally yield more reliable estimates of hydroclimate variability.



Our study did not explicitly aim to compare hydroclimate variability in the reconstruction with that of climate model simulations. However, initial analyses reveal a more pronounced multi-decadal to centennial hydroclimate variability in the reconstruction, corroborating previous model-data comparisons for speleothems and isotope-enabled climate models. For tropical South America, (Orrison et al., 2022) demonstrated that the GISS and iCESM last millennium simulations underestimate centennial monsoon intensity changes in the isotopic composition of precipitation. On a global scale, (Bühler et al., 2020) found an underestimation of multi-decadal to centennial variability, using climate simulations also included in our study. A similar underestimation of multi-decadal to centennial variability was previously found for precipitation in the CMIP5 Last Millennium ensemble both for tropical South America (Parsons et al., 2018) and globally (Ault et al., 2012; Parsons et al., 2017), where the spectra were shown to resemble that of white noise. Additionally, for temperature, it is now established that while climate models generally capture the climate variability of global mean temperature, they underestimate regional temperature variability (Laepple et al., 2023b). Our PaleoDA reconstruction, thus, represents a compromise between climate models and climate proxy records, potentially providing a more realistic representation of hydroclimate variability scaling than models alone. We emphasize that the reconstruction of multi-decadal to centennial variability would not have been feasible without the multi-timescale PaleoDA approach, enabling the inclusion of speleothems primarily for technical reasons. It remains to be determined if the use of multi-year covariances, instead of annual ones, in the PaleoDA algorithm also contributed to this, a matter beyond the scope of this publication but suggested by pseudo-proxy experiments (Steiger and Hakim, 2016; Choblet et al., 2023).

In principle, our reconstruction could be employed to investigate interannual variations in SASM strength, for which ENSO is considered a driver (Garreaud et al., 2009). However, we only employed the key archive for SASM variability, the speleothems, at a decadal resolution, even though many of the records technically have a finer resolution. The conservative approach of using decadal time scales for speleothems was adopted to avoid additional complexity arising from dating uncertainty, sampling from age-model ensembles during the reconstruction and taking into account the transit time of cave seepage water. Developing a scheme that considers all these factors would enable to study the influence of ENSO on SASM variability. Employing speleothems at higher resolutions might also improve validation results against 20th-century instrumental precipitation in the region.

### 6.4 Future developments

To enhance the PaleoDA reconstruction, exploring more elaborate PSMs could be beneficial to perform absolute value reconstructions instead of anomaly reconstructions. In previous work leading up to this study, the speleothem PSM proposed by Dee et al. (2015) was tested, but did not reduce offsets between models proxy records. This can result from either model-biases, from PSMs that do not incorporate processes known to a local cave expert, or from lack of physical understanding of the records. Therefore, we compared simulation and proxy record anomalies by taking known seasonality into account or via a linear regression for tree and coral time series. For speleothems, including information from local cave monitoring studies (e.g. Moquet et al., 2016; Sekhon et al., 2021; Jiménez-Iñiguez et al., 2022), could provide a better understanding of the proxy record and help recognize model biases in simulated $\delta^{18}O$ of precipitation values as well as improve PSMs. However, this usually



requires more metadata of individual cave systems, which is not always available and may reduce the selection of usable proxy records.

When employing more elaborate PSMs in PaleoDA, it is essential to test their effect on the covariance patterns as nonlin-
ear mathematical operations can impact covariances unexpectedly. Applying PSMs of higher complexity to instrumental data could also improve the assessment of proxy error and SNR. Although we consider the employed SNR of 0.5 ratio reasonable, it remains a broad assumption due to the lack of sufficiently long instrumental data at the cave locations. Previous PaleoDA reconstructions without clear proxy record error have used a post-hoc adjustment of the error variance (Tierney et al., 2020; Osman et al., 2021). However, we refrain from doing so here, as the tuned errors might be confounded by model biases in the covariances and the proxy record error should represent a real physical and statistical property rather than a tunable hyperpa-
rameter. An alternative reconstruction setting proxy record variance equal to the prior record variance yielded similar climate fields with more pronounced magnitudes in hydroclimatic changes (AF4). Due to the similarity, we have not investigated these reconstructions further, but emphasize, that in the light of the unclear SNR of speleothems and other proxy records, an equal prior/proxy weighting approach would be equally legitimate.

## 7 Conclusion

This study presents the first South American climate field reconstruction for the entire CE and the entire continent. By incor-
porating speleothems and other non-annually resolved climate archives through a multi-timescale Data Assimilation approach, we have improved spatial proxy record coverage for South America compared to previous global reconstructions and elimi-
nated the need for uncertain calibration of low-resolution isotopic records to temperature and precipitation. Our primary focus was centered on centennial climate changes, with particular attention to the intensity of the SASM. The reconstruction reveals compelling evidence of a strengthened monsoon during the LIA, contrasting with a weaker SASM before, in particular during the early phase of the MCA. We also found a strengthening during the first centuries of the CE, which remains more elusive due to the more limited number of proxy records. In our reconstruction, speleothems played a vital role in capturing centennial variability.

While this study showcases the potential of the multi-timescale PaleoDA reconstruction approach including speleothems, we acknowledge certain limitations that warrant further investigation. The reconstruction's validation, uncertainties in proxy record errors, spatial smoothing of the reconstruction, and the necessity for a larger ensemble of isotope-enabled climate model simulations with diverse models are important areas for future research. Additionally, exploring differences in simulated co-
variance patterns, fundamental to the PaleoDA EnKF reconstruction algorithm, holds promise for refining and enhancing this methodological approach. We make our reconstructed datasets publicly available, providing a foundation for future climato-
logical data analysis including comparisons to proxy records, other reconstructions, and climate simulations. Furthermore, by publishing the code for the multi-timescale PaleoDA algorithm alongside this study, we encourage and enable the application of the multi-timescale PaleoDA method to proxy records of different temporal resolutions. This concept has great potential for



global climate field reconstructions, particularly for the CE and older time periods, where speleothems and other non-annually

and irregularly resolved proxy records serve as essential indicators of past hydroclimate changes.

*Code and data availability.* Code to reproduce reconstructions, figures and data preprocessing are available on GitHub https://github.com/ mchoblet/paleoda_sa. The reconstructed climate fields are available via a Zenodo repository https://zenodo.org/records/10622265. We recommend using the multi-model ensemble mean of the reconstructions. The input data for running the reconstruction is accessible via a second Zenodo repository https://zenodo.org/records/10370001. We use the model data originally available at https://zenodo.org/records/7516327

(Bühler et al., 2022).

The SISAL (Speleothem Isotopes Synthesis and AnaLysis Working Group) version 3 database (SISALv3) is publicly available at https://doi.org/10.5287/ora-mzy8pozvk (Kaushal et al., 2023). The PAGES2k Database (Emile-Geay et al., 2017) is available at https://lipdverse.org/project/pages2k/. The Iso2K Database (Konecky et al., 2020) is available at https://lipdverse.org/project/iso2k/. The South American Drought Atlas Database is available at https://www.cr2.cl/datos-dendro-sada/. References for the individual records are listed in Appendix

D.

The PaleoDA code and the figures were created using the Python programming language (Van Rossum and De Boer, 1991), version 3.10 and with a collection of open-source packages, notably Xarray (Hoyer and Hamman, 2017), Numpy (Harris et al., 2020a), Matplotlib (Hunter, 2007), Cartopy (Met Office, 2010 - 2015), Pyleoclim (Khider et al., 2022), Scipy (Virtanen et al., 2020), Pandas (Wes McKinney, 2010) and Numba (Lam et al., 2015).

*Video supplement.* The reconstructed climate fields are also provided for the *all proxy records* reconstruction in animated form for

– Temperature, Precipitation, SPEI and $\delta^{18}$O climate fields: https://av.tib.eu/media/66877 (DOI: 10.5446/66877).

– Precipitation reconstruction and monsoon curve side by side, comparison to the reconstruction by Neukom et al. (2019) https://av.tib.eu/media/66879 (DOI: 10.5446/66879).

– Precipitation reconstruction and monsoon curve side by side, overlaying of the speleothem $\delta^{18}$O anomalies, comparison to the recon-

struction by Neukom et al. (2019) https://av.tib.eu/media/66880 (DOI: 10.5446/66880).

*Author contributions.* MC, JB, VN, NS and KR designed this study. MC developed the PaleoDA code and produced the reconstructions. MC and JB wrote the paper and MC created the figures. VN, NS and KR contributed with data interpretations and to the revisions of the manuscript. All authors approved of the final version of the paper.

*Competing interests.* The authors declare that they have no conflict of interest.



*Acknowledgements.* As this study includes data complied by SISAL (Speleothem Isotopes Synthesis and Analysis), we thank the Pages2k and the Iso2k network, working groups of the Past Global Changes (PAGES) project. We thank all initial authors that provided model simulation data, proxy record data and those researchers who compiled the proxy record databases. We thank Mariano Morales from CONICET, Argentina for sharing tree ring data from the South American Drought Atlas and further tree ring collections and Michael Erb and Matt Osman for fruitful discussions about PaleoDA. Nils Weitzel, Beatrice Ellerhoff and Markus Maisch are acknowledged for helpful advice and
comments during the elaboration of this project.

*Financial support.* This research has been supported by the Deutsche Forschungsgemeinschaft (grant nos. 316076679, 395588486, and 442926051) and the Bundesministerium für Bildung und Forschung through the PalMod project (grant no. 01LP1926C). NS is supported by the Israel Science Foundation (grant no. 2654/20).



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





# Appendix A: Additional Figures

**Detailed spatiotemporal availability**

**AF1. Spatiotemporal availability of the proxy records for each century** . The black box represents the core region of the South American Summer Monsoon (Vuille et al., 2012)



**Algorithm sketch**

## Spatial information

**Climate Model Data from Last Millennium simulations for**

- $\delta^{18}$O
- Temperature
- Precipitation
- SPEI
- Monsoon Index

## Temporal information

**Proxy record data**

**Proxy System Model (PSM) / Calibration to instrumental data**

**Observation estimates using PSM/Calibration parameters**

**Prior state matrix**

**Debias**

**Resample to target resolution**

**Estimate Proxy Error Variance**

**Ensemble Kalman Filter equations (Offline)** time

**Reconstruction**

**AF2.** Algorithm sketch for multi-timescale Paleoclimate Data Assimilation Algorithm. See main text for a description of the individual steps.




1110 **Mean anomaly fields (Figure 3) for austral summer reconstruction**

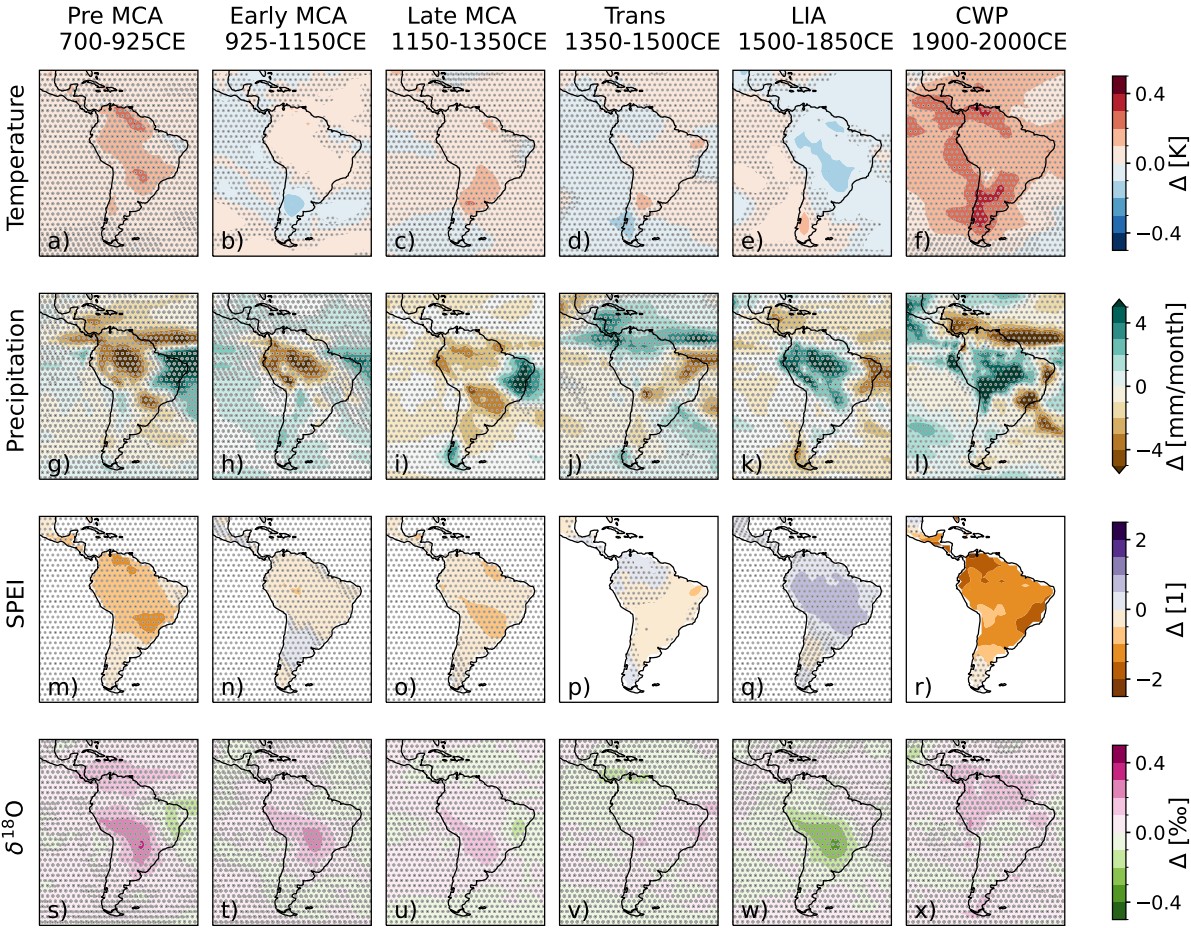

**AF3.** Same as Figure 3 for the austral summer (DJF) reconstruction. Stippling indicates grid cells where the difference to the Last Millennium values is not significant according to a Welch's t-test ($\alpha > 0.01$).





**Mean anomaly fields (Figure 3) for reconstruction with different proxy error definition**

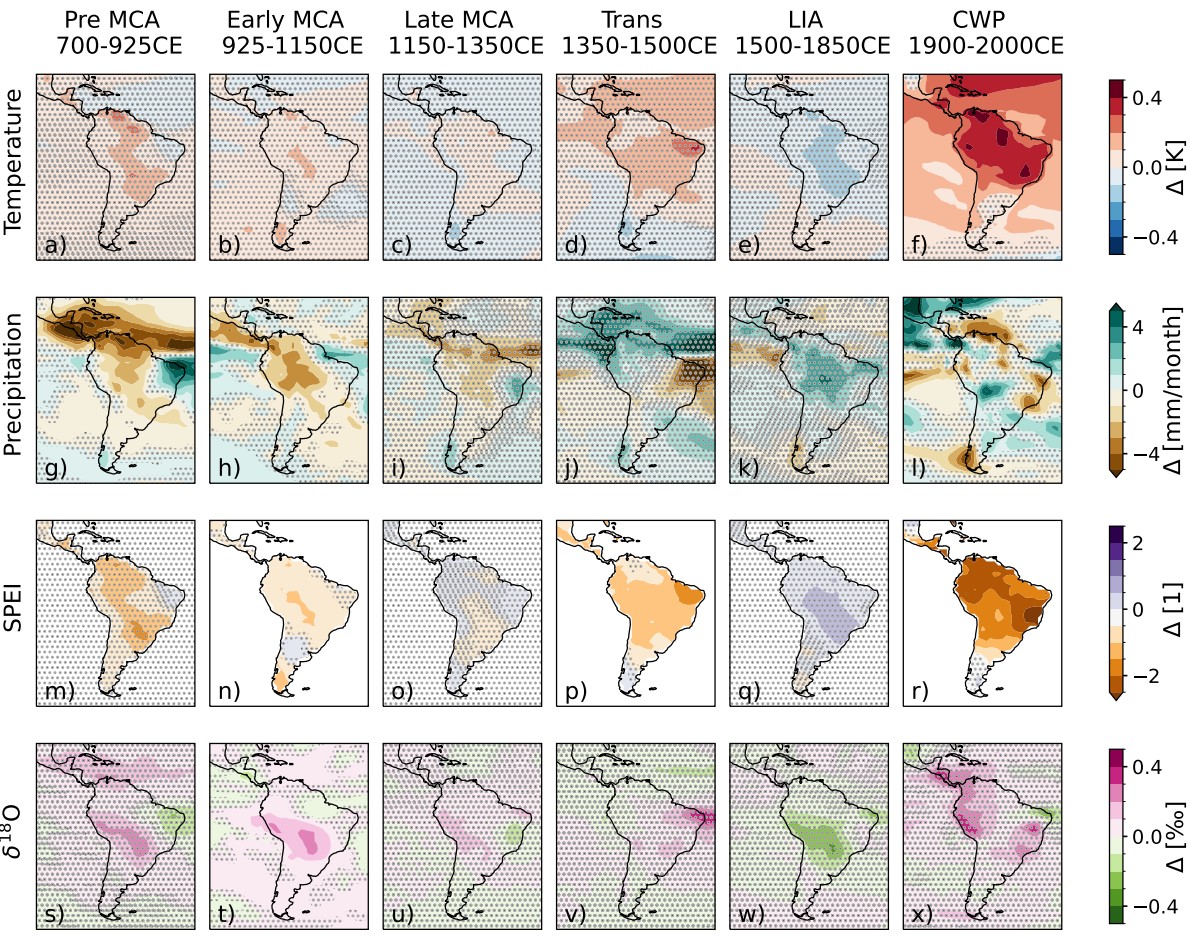

**AF4.** Same as Figure 3 applying a proxy error variance equal to the prior variance instead of the SNR=0.5 proxy error definition. Stippling indicates grid cells where the difference to the Last Millennium values is not significant according to a Welch's t-test ($\alpha > 0.01$).





**Centennial climate anomalies**

## Centennial precipitation anomalies wrt 850-1850 CE



**AF5.** Reconstructed mean anomaly fields for precipitation with respect to the 850 - 1850 CE mean value (annual reconstruction). Stippling indicates grid cells where the difference to the Last Millennium values is not significant according to a Welch's t-test ($\alpha > 0.01$).





# Centennial temperature anomalies
# wrt 850-1850 CE

**AF6.** Reconstructed mean anomaly fields for surface temperature with respect to the 850 - 1850 CE mean value (annual reconstruction).

Stippling indicates grid cells where the difference to the Last Millennium values is not significant according to a Welch's t-test ($\alpha > 0.01$).



**AF7.** Reconstructed mean anomaly fields for SPEI with respect to the 850 - 1850 CE mean value (annual reconstruction). Before computing centennial means, the SPEI time series have been standardized with respect to the 850 - 1850 CE mean. Stippling indicates grid cells where the difference to the Last Millennium values is not significant according to a Welch's t-test ($\alpha > 0.01$).









**AF8.** Reconstructed mean anomaly fields for $\delta^{18}$O with respect to the 850 - 1850 CE mean value (annual reconstruction). Stippling indicates grid cells where the difference to the Last Millennium values is not significant according to a Welch's t-test ($\alpha > 0.01$).



**Mean anomaly fields (Figure 3) for reconstruction only using speleothem proxy records**

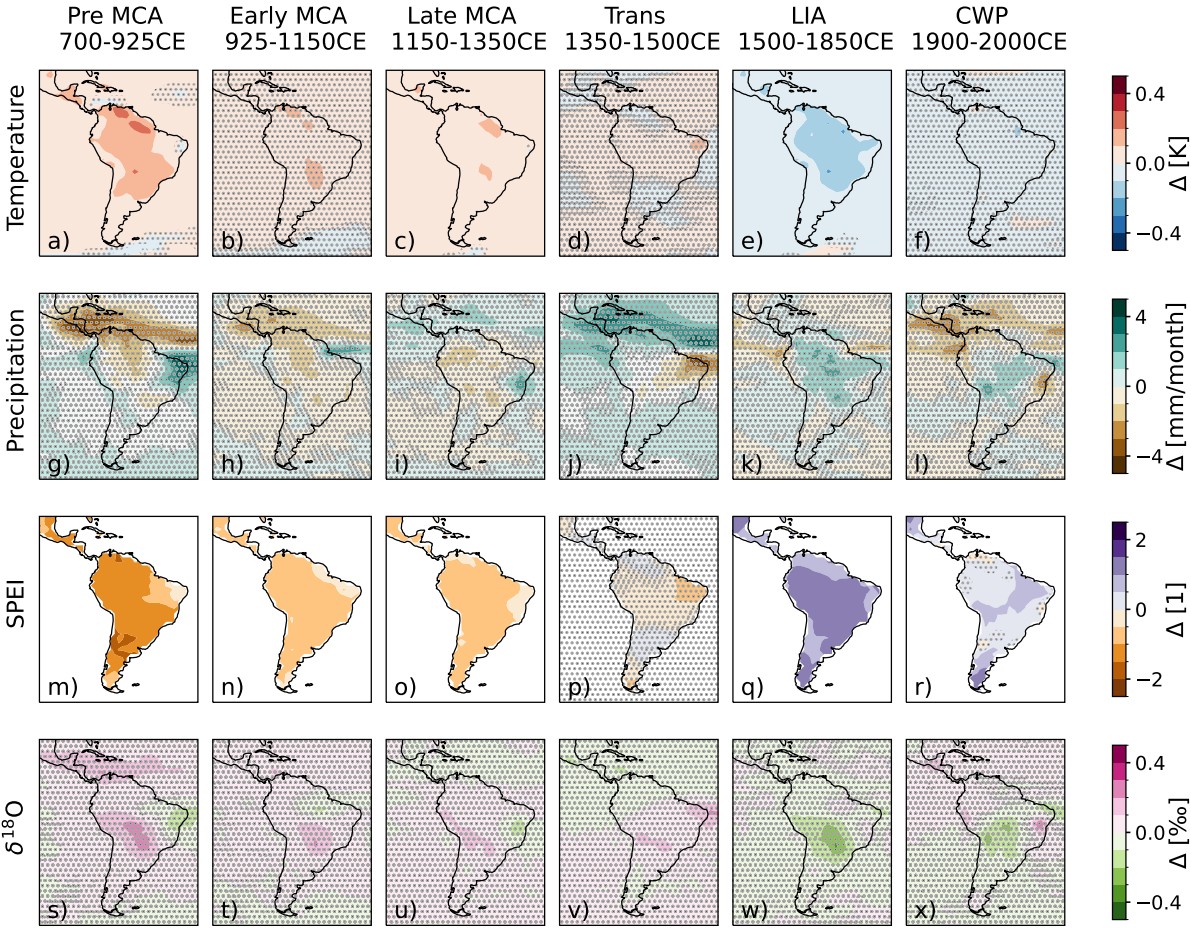

**AF9.** Same as Figure 3 but for the reconstruction that only uses speleothems as proxy record input data. Stippling indicates grid cells where the difference to the Last Millennium values is not significant according to a Welch's t-test ($\alpha > 0.01$).





**Mean anomaly fields (Figure 3) for reconstruction excluding speleothem proxy records**

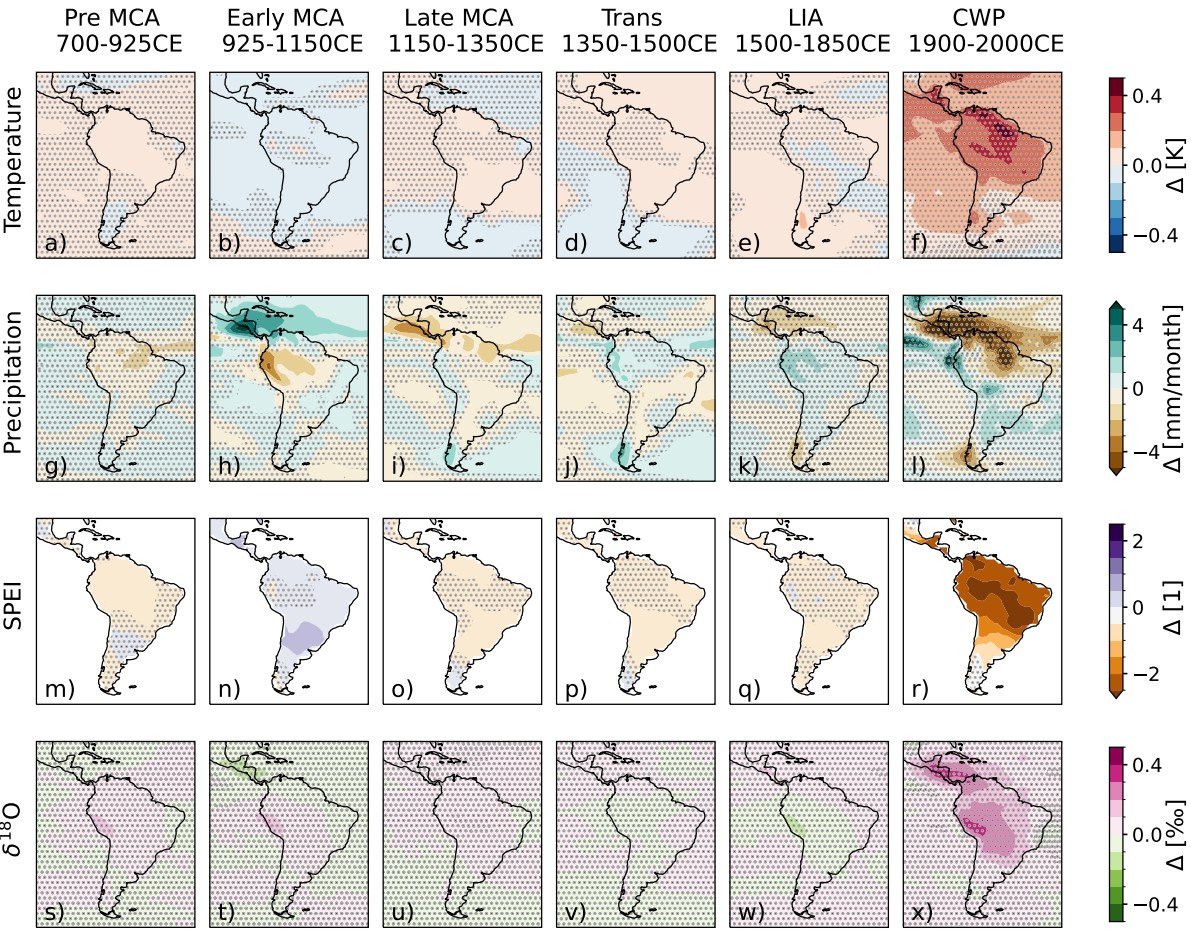

**AF10.** Same as Figure 3 but for the reconstruction that uses all proxy records except the speleothems as proxy record input data. Stippling indicates grid cells where the difference to the Last Millennium values is not significant according to a Welch's t-test ($\alpha > 0.01$).





1115  **Additional SASM index figures**

**AF11.** Same as Figure 4 for the austral summer (DJF) reconstruction with extended y-axes ranges. The PHYDA reconstruction displayed here is a specific austral summer reconstruction, whereas LMRv2.1 is only provided at an annual time scale.







**AF12.** Same as Figure 4 including the single model reconstructions using all proxy records (dotted lines). The black line is the multi-model ensemble reconstruction (mean of single prior reconstructions).





**AF13.** Same as Figure 4 for the *all proxies* reconstruction and the model simulations. The time period has been limited to the period 850-1850 CE as this is the time span covered by the model simulations.



**Additional power spectra**

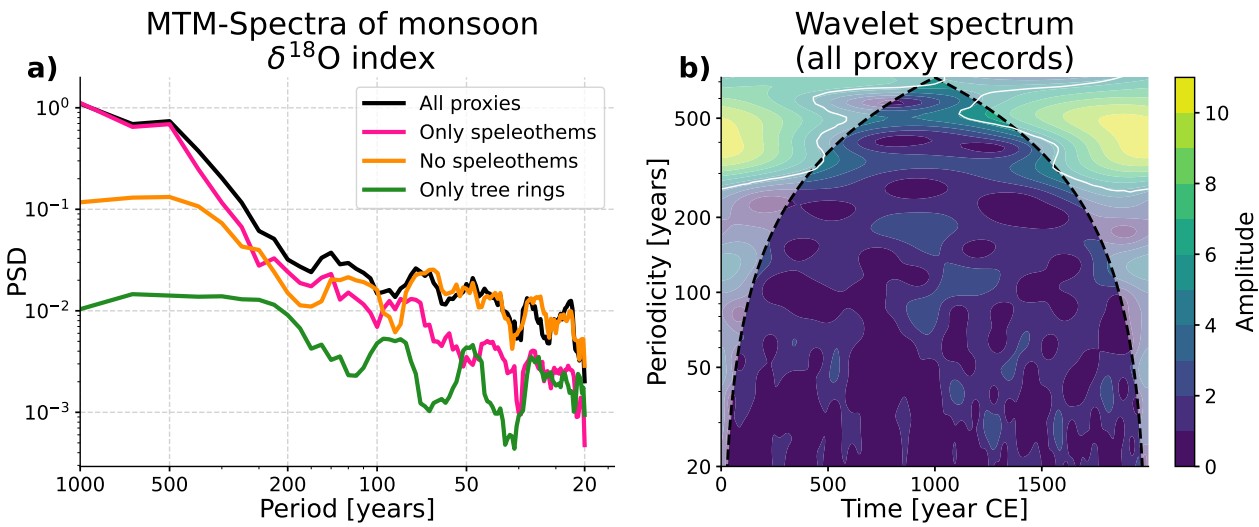

**AF14. Spectra of reconstructed monsoon $\delta^{18}$O index**. See description of Figure 5.

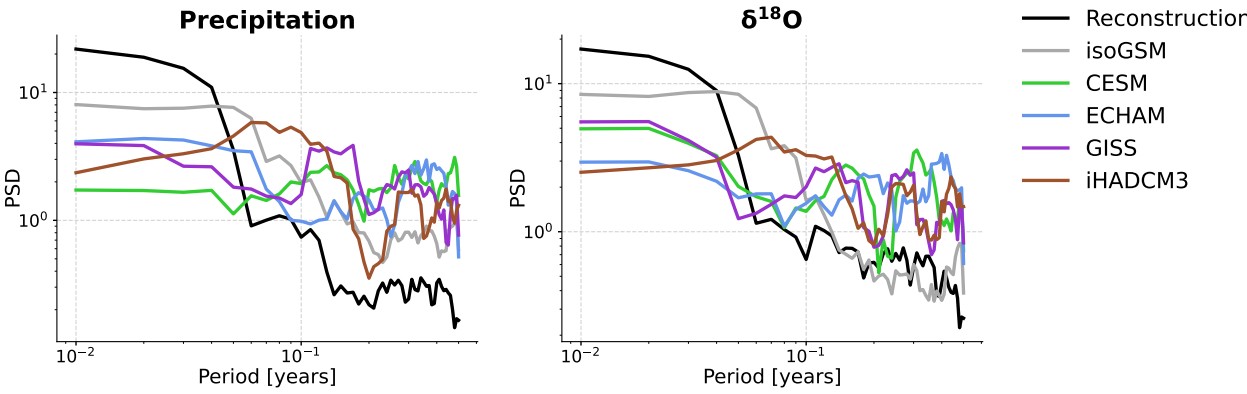

**AF15. Spectra of SASM indices in model simulations**. The spectra for SASM precipitation and $\delta^{18}$O have been computed as in Figure 5, but limited to the period 850-1850 CE, because this is the time period covered by the model simulations. In addition, all time series have been standardized, as the model simulations have more overall variability (higher variance). Standardizing the time series allows to highlight the different scaling of the simulation and reconstruction spectra.





## Climate anomalies in PHYDA and LMRv2.1

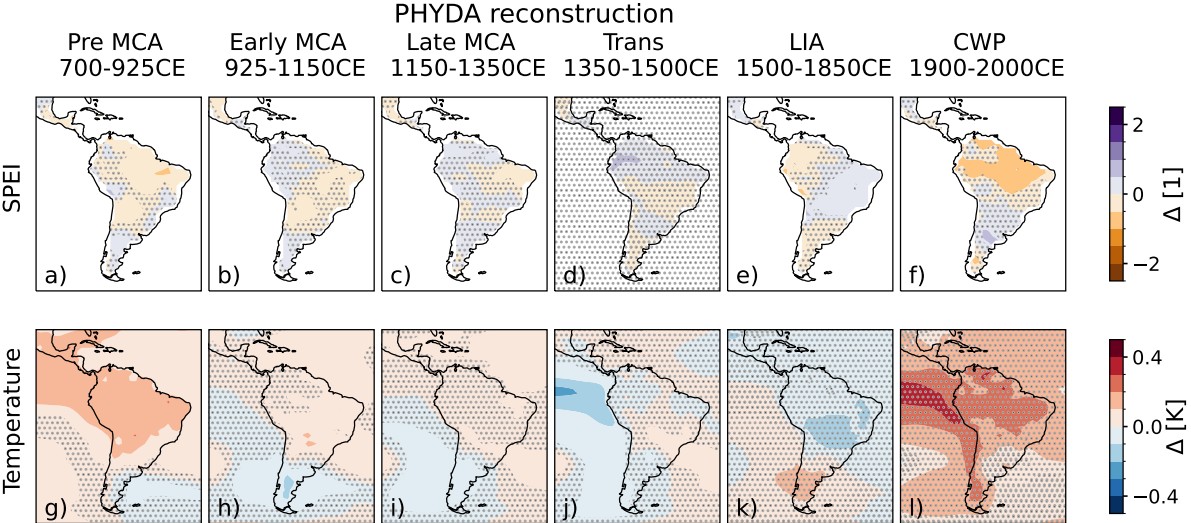

**AF16.** Same Anomaly Fields as Figure 3 for the PHYDA reconstruction (Steiger et al., 2018), which includes temperature and SPEI among its reconstructed variables, but not precipitation and $\delta^{18}$O. Stippling indicates grid cells where the difference to the Last Millennium values is not significant according to a Welch's t-test ($\alpha > 0.01$).

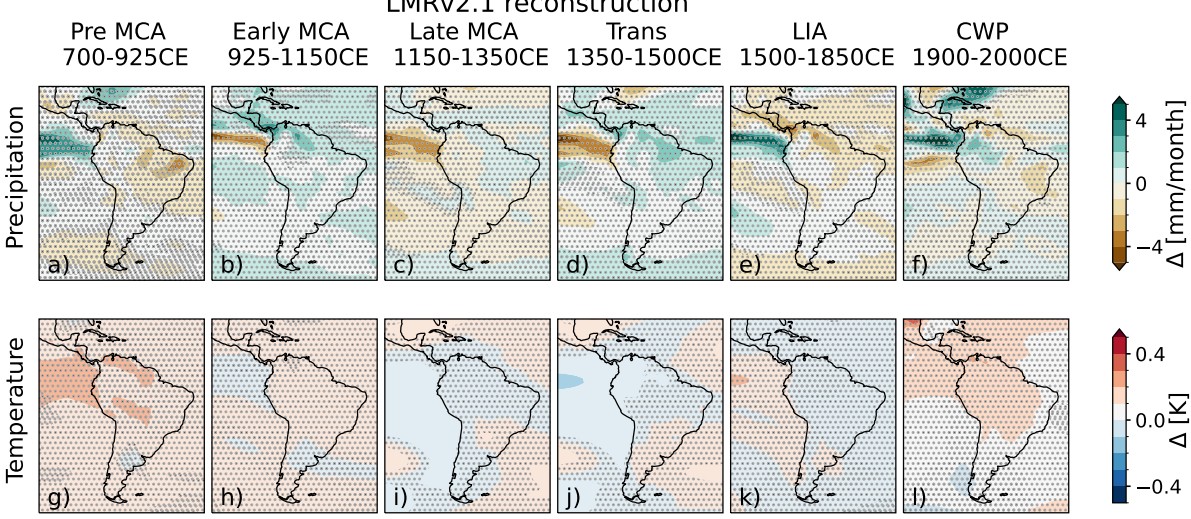

**AF17.** Same Anomaly Fields as Figure 3 for the LMRv2.1 reconstruction (Tardif et al., 2019), which includes temperature and precipitation among its reconstructed variables, but not SPEI and $\delta^{18}$O. Stippling indicates grid cells where the difference to the Last Millennium values is not significant according to a Welch's t-test ($\alpha > 0.01$).





**Correlations in the model simulations (prior) for NEB**

**AF18. Correlation of** $\delta^{18}$**O mean in the Nordeste** (black box, Lat: -15-0°, Lon: 313-327°) to climate variables $\delta^{18}$O, precipitation, temperature and SPEI of individual grid cells in the five isotope-enabled climate model simulations. The correlations have been computed with the annual mean values of the simulated climate variables.



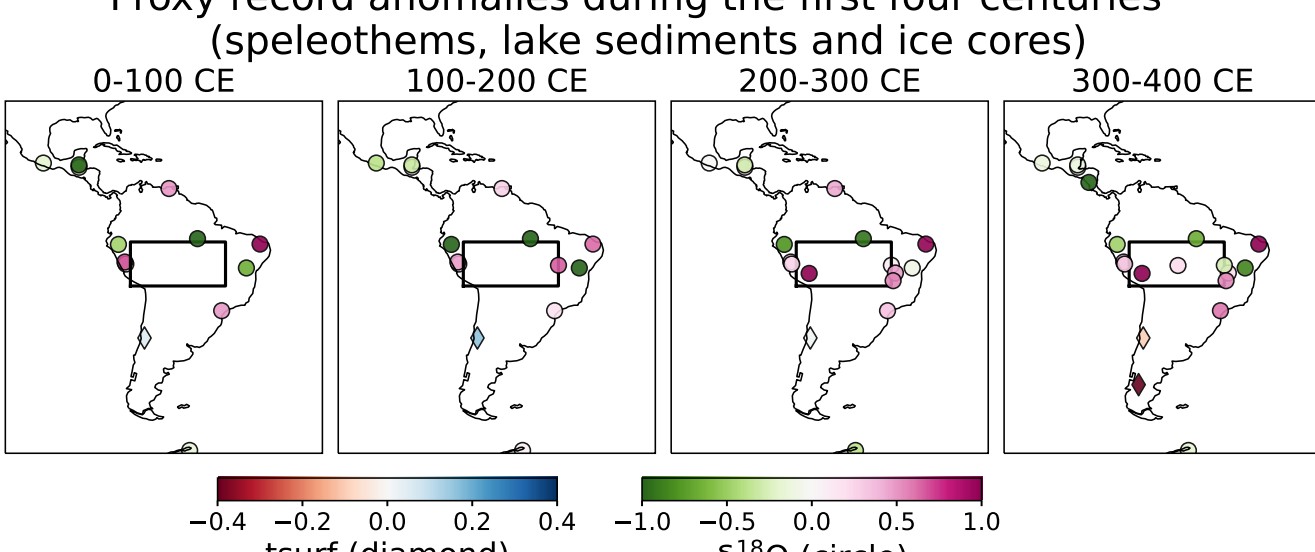

**AF19.** Proxy record anomalies during the first four centuries of the CE with respect to the Last Millennium mean.



# Correlation to SASM precipitation

**AF20. Correlation of precipitation mean in core SASM region** (black box, Lat: -17.5 - -5°, Lon: 287.5-312.5°) to climate variables $\delta^{18}$O, precipitation, temperature and SPEI of individual grid cells in the five isotope-enabled climate model simulations. The correlations have been computed with the annual mean values of the simulated climate variables.



## Appendix B: Validation of reconstruction

### B1 Spatial reconstruction skill using gridded instrumental data

To evaluate the spatial reconstruction skill of our reconstructions, we compare these to gridded instrumental datasets, Berkeley Earth (Rohde and Hausfather, 2020) for temperature and CRUTS 4 (Harris et al., 2020b) for precipitation. A reference SPEI dataset is computed from CRUTS 4 using Thornthwaite's method for estimating potential evapotranspiration. The validation periods are 1920 - 2000 CE for temperature and 1950 - 2000 CE for precipitation and the SPEI, due to limited local instrumental, radiosonde and satellite precipitation data during the first half of the 20th century. Until the second half of the 20th century, South America lacked extensive weather station coverage, with only a few stations mainly situated in coastal regions and the Southern part of the continent (See Figure 1 in Harris et al., 2020b). Estimates of precipitation, which is a more localized phenomenon than temperature, are particularly affected by this limited station coverage. Precipitation estimates are considered more reliable since the deployment of the global radiosonde network in 1958 and satellite-derived rainfall estimates from the late 1970s (Garreaud et al., 2009). Consequently, the time period suitable for meaningful calibrations/validations using instrumental precipitation data is usually restricted to the second half of the 20th century (e.g. Morales et al., 2020). The validation period is, thus, the same as the calibration period of the tree rings and corals, which can be considered problematic. Unlike other climate field reconstruction techniques such as PCR (e.g. Neukom et al., 2010, 2011), PaleoDA with linear statistical PSMs does not strictly require the separation of calibration/validation data for two reasons. Firstly, the statistical PSM estimates regression parameters using local observational data, which are then applied to independent model simulation data. The model simulation data's mean and spatial covariances are not debiased with respect to the observational data, and thus importantly influence the reconstruction. In contrast, techniques like Point by Point regression (e.g. Morales et al., 2020) estimate a linear regression model for each reconstructed grid cell, resulting in more instrumentally tuned reconstructions. Secondly, the calibration process predicts proxy records from instrumental data, while the validation process predicts instrumental data from proxy records. This non-symmetry introduces information loss. Additionally, using multiple variables as predictors simultaneously in the reconstruction can potentially introduce errors through inadequate covariances in the prior.

As skill metrics, we choose the widely employed Pearson correlation and the Continuous Ranked Probability Skill Score (CRPSS) (Wilks, 2011). The correlation is a simple similarity metric with range $[-1, 1]$, which rewards a correct phasing of the reconstructed signal with respect to the observational data. In the correlation plots, we further denote significant correlations according to an effective p-value $< 0.05$. The effective p-value takes into account the smaller number of degrees of freedom in autocorrelated time series (Bretherton et al., 1999). Note, that, in a strict sense, correlations require detrended time series. We choose to not detrend the time series to also evaluate the skill of the reconstruction to capture trends, similar to previous PaleoDA reconstructions (e.g. Tardif et al., 2019; Steiger et al., 2018) In contrast, the CRPSS is considered a strictly proper scoring metric (Gneiting and Raftery, 2007) because it takes into account the posterior reconstruction distribution instead of the ensemble mean. We assume Gaussian statistics and, thus, characterize it via the ensemble mean and standard deviation. The CRPSS is the skill score version of the Continuous Ranked Probability Score (CRPS) and is computed as 1-CRPS$_{rec}$/CRPS$_{ref}$, where CRPS$_{ref}$ is the CRPS value for a reference distribution. CRPS rewards small biases, correct variances and ensemble



spread. For the reference CRPS score, we take the ensemble statistics of the uninformed prior ensemble as in Steiger et al.

(2018). The CRPSS values lie in the range $(-\infty, 1]$, where positive values denote reconstructions more skillful than the prior.

As the CRPSS values are computed for each validation time step, we compute the temporal mean of these values and denote

them as CRPSS. Both metrics do not take observational uncertainties into account. For all skill scores, we have initially also

considered applying a low-pass filter to the time series before applying the skill metrics, as we estimate our reconstruction to be

more meaningful on longer than annual time scales. However, we have finally refrained from this idea as the validation period

is pretty short (80 and 50 years) and for correlation, higher absolute correlations and effective p-values are expected due to the

filtering introducing auto-correlation.

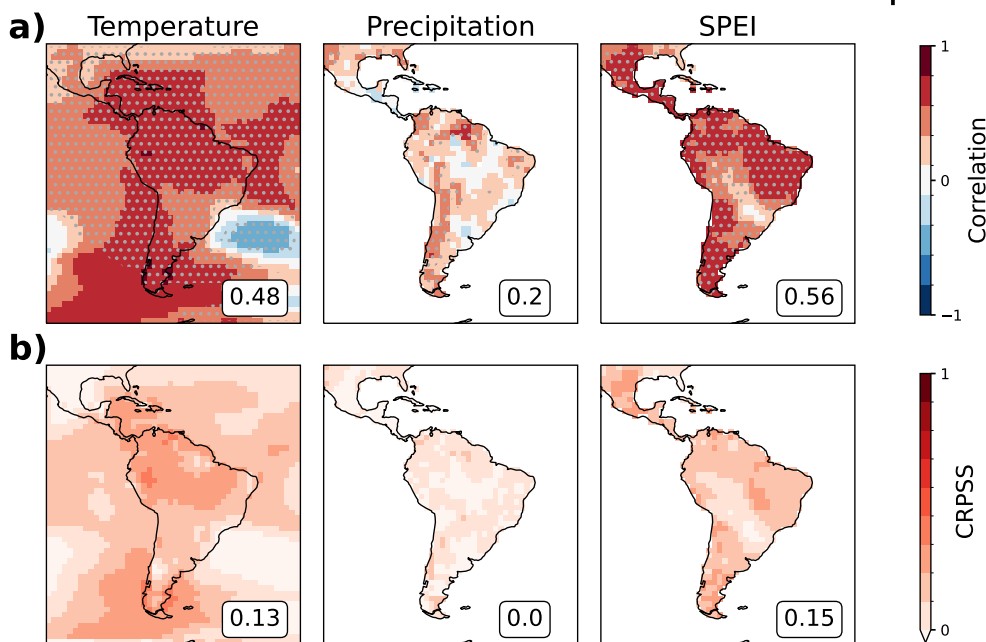

**AF B1. Comparison to validation data sets over calibration period:** validation skill metrics from comparing our reconstruction to the
Berkeley Earth surface temperature dataset (Rohde and Hausfather, 2020) for the years 1920 - 2000 CE, the CRUTS 4 precipitation dataset
(Harris et al., 2020b) for the years 1950 - 2000 CE and SPEI calculated using temperature and precipitation from the CRUTS 4 dataset for
the years 1950 - 2000 CE. Precipitation skill is only evaluated on land as the instrumental dataset only provides precipitation over land.
The panels in a) show the correlation with stippling indicating effective p-values smaller than 0.05. The panels in b) show the results for
the CRPSS. The values in the lower right corner denote the mean skill score. See text for details. The spatial reconstruction skill for the
reconstruction of austral summer (DJF) is displayed in Figure AF B2.





## Comparison to validation datasets over calibration period (DJF)

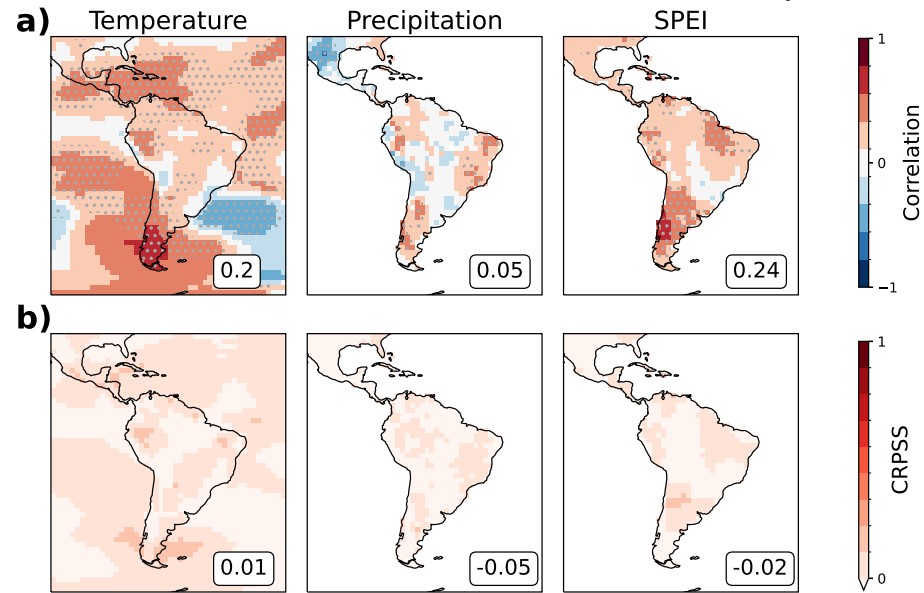

**AF B2. Validation skill metrics** from comparing our austral summer (DJF) reconstruction to instrumental data. See caption of Figure AF B1 for details.

Figure AF B1 shows the skill metric results for the annual reconstruction. Significant positive correlations are found for almost all terrestrial locations in the surface temperature and SPEI reconstruction (mean values 0.48 and 0.56), whereas for precipitation positive significant correlation is only found in the Andes and parts of northern and eastern South America (mean value 0.20) Due to the increasing temperature trend of the current warm period (CWP), such high correlations can be expected, also for SPEI, which depends on temperature via evapotranspiration. For precipitation, we find the highest similarities close to the proxy record locations, in particular the Andes, which reflects the local character of precipitation changes and the lack of a clear trend in precipitation during the 20th century. CRPSS values for temperature and SPEI are mostly positive (mean 0.13 and 0.15), whereas for precipitation positive values stay close to zero (mean 0.00).

The skill metric values for the austral summer reconstruction (DJF) are shown in Figure AF B2. The values are consistently lower than for the annual means, with the highest similarity encountered for the Southern Cone. We suppose, that despite calibrating the tree rings and corals to austral summer means, their proxy record locations are less representative for the whole continent compared to annual means. In the presentation of the results, we thus focus on the annual reconstruction.

Note, that this type of 20th century validation mainly reflects the skill of reconstructing the continent's hydroclimate through tree proxy data, the most abundant annually dated climate archive in our database during the instrumental era. The speleothems, which are a key archive for reconstructing the entire past two millennia, are only used on a decadal time scale and thus only contribute very little to reconstructions during the 20th century. Moreover, the focus on our data analysis will lie on decadal to centennial climate changes. The instrumental record is too short to validate the reconstructions skill on these time scales.



Finally, it needs to be accounted that in most climate field reconstructions, the number of proxy records available in the validation/calibration period is usually order of magnitudes larger than in the preceding centuries. Hence, the obtained skill is not necessarily representative for the rest of the reconstruction period.

## B2 Precipitation in the core monsoon region

A central aspect of this study consists of reconstructing the variability of the SASM according to the definition of Vuille et al.
(2012), who proposed computing the mean precipitation in the core monsoon region (5° S–17.5° S/72.5° W–47.5° W, see black rectangle in Figure 1) as an indicator of monsoon strength. The SASM reaches its peak intensity during the austral summer months (DJF). Nevertheless, we focus on annual mean precipitation values in the core monsoon region for the validation, as we consider our annual reconstruction to be more reliable (see Section B1). Figure AF B3 shows the reconstructed mean precipitation anomaly in the core monsoon region compared to the values computed from the instrumental datasets CRUTS
4 (Harris et al., 2020b) and GPCC (Schneider et al., 2008) for the 20th century. It is apparent that our reconstruction does not capture short-term precipitation fluctuations and clearly underestimates precipitation variability for the short instrumental period, as it stays in the range of [-5,5] mm/month, whereas the instrumental data fluctuates in the range [-20,20] mm/month. The intensity of these fluctuations also shows considerable disagreement between the two instrumental datasets during the first half of the 20th century.

Table B1 presents the skill scores for the reconstructed monsoon index depending on whether all proxy records, only tree data, or all proxy records without the tree data is used. These scores have been computed for the calibration period, but also the entire 20th century. Overall, the skill scores are low and inconclusive. In contrast to what one would expect, the tree data, which is calibrated to instrumental temperature, precipitation, and SPEI data, yields the lowest skill scores. The tree data, which is mainly located in the central and southern Andes and thus outside the core monsoon region does not seem to be a good
predictor for mean precipitation changes in the core monsoon region. This emphasizes the need to add precipitation sensitive proxies within, or at least closer to the core monsoon region. In our study, additional value is gained through the inclusion of speleothem records, but which are not highly resolved enough for a validation during the short instrumental period. This assessment emphasizes that the reconstructed precipitation changes likely do not accurately capture short-term variations in precipitation. We assume that the recorded changes are more likely to reflect long-term trends, which cannot be adequately
validated due to the limited duration of instrumental data. Due to its limitations and despite its frequent use in PaleoDA, the instrumental validation exercise is thus not a proper tool for our reconstruction.

## B3 Drought Index validation for the Southern Cone

A reference reconstruction of dry and wet conditions for Southern South America in the period 1400 - 2000 CE is the South American Drought Atlas (SADA) (Morales et al., 2020), whose input tree ring data is also partially employed our study.
The SADA reconstructs the self-calibrated PDSI (scPDSI) using Point-by-Point Regression, a climate field reconstruction technique that is more calibrated towards local instrumental data. The SADA can be considered the most elaborate drought index reconstruction for the region. Although our reconstruction does not include scPDSI as a reconstructed variable, we

 

|  |  | 1950 - 2000 CE |  | 1901 - 2000 CE |  |
| --- | --- | --- | --- | --- | --- |
|  |  | CRUTS 4 | GPCC | CRUTS 4 | GPCC |
| All proxy records | Corr | 0.16 | 0.03 | 0.08 | 0.01 |
|  | CRPSS | -0.01 | -0.04 | -0.04 | -0.03 |
| Only trees | Corr | -0.06 | -0.14 | -0.10 | -0.08 |
|  | CRPSS | -0.07 | -0.06 | -0.09 | -0.05 |
| No trees | Corr | 0.18 | 0.10 | 0.15 | 0.08 |
|  | CRPSS | -0.05 | -0.04 | -0.07 | -0.03 |

**Table B1.** Skill scores for the annual reconstruction of precipitation in the core monsoon region of the SASM during the 20th century with respect to CRUTS 4 (Harris et al., 2020b) and GPCC (Schneider et al., 2008). For the experiments of the first row, all proxy records have been used, in the second row only tree data has been employed and in the third row the tree data has been explicitly excluded. The correlation values are presented without effective p-values, as none was significant ($\alpha < 0.05$).

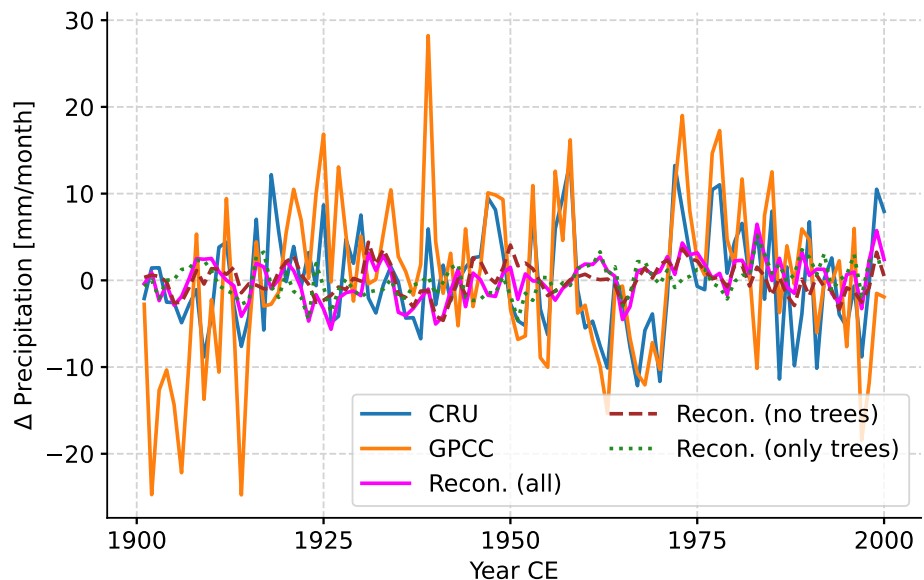

**AF B3. Annual precipitation anomaly in the core monsoon region** ($5° - 17.5°$ S / $72.5° - 47.5°$ W) for the years 1901 - 2000 CE in the CRUTS 4 and GPCC datasets and our reconstruction (including all proxy records). For all time series the mean of the period 1901 - 2000 CE was subtracted.

compute the correlation to our reconstructed SPEI drought index to assess their similarity (Figure AF B3). Despite scPDSI and PDSI capturing different types of droughts, we expect similarity in the phasing of drier and wetter periods due to using similar input data. Additionally, the reconstructed SPEI is compared to the SPEI from the PHYDA reconstruction (Steiger et al., 2018), which also employs PaleoDA as a reconstruction technique using a subset of the SADA tree data. In PHYDA, where both SPEI





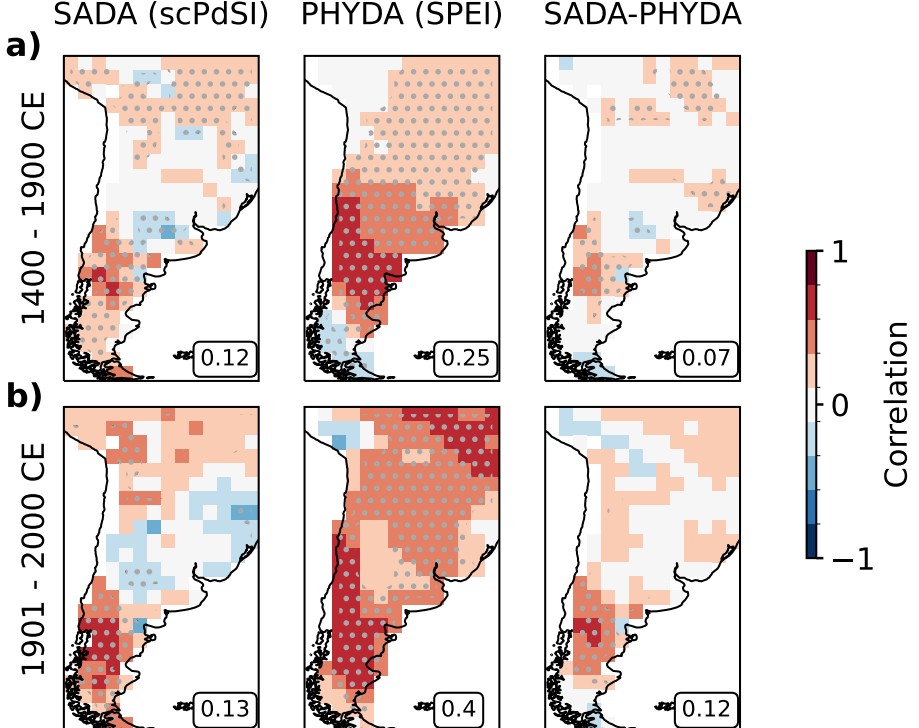

**AF B4.** Evaluation of correlation of the reconstructed annual SPEI to scPDSI from the SADA (Morales et al., 2020) and SPEI from the PHYDA (Steiger et al., 2018) (left and center panels). The right panel compares the correlation of scPDSI in SADA and PDSI in PHYDA. Grid cells with effective p-values <0.05 are indicated by stippling. The upper row (a) shows the correlation for the period 1400 - 1900 CE and the lower row (b) for the period 1901 - 2000 CE. The SADA and PHYDA datasets have been regridded to the spatial resolution of our reconstruction.

and PDSI are reconstructed, both indices are highly correlated (not shown), we thus consider that in PHYDA both indices can be used interchangeably.

The largest similarity in terms of significant positive correlation for our reconstruction and SADA can be found close to
the tree data locations in the Andes, especially Patagonia. The lowest similarity is found in the Pampas and the La Plata basin. Comparing the similarity of our reconstruction to SADA to the similarity of SADA and PHYDA, we find higher mean correlations than for PHYDA (0.12 vs 0.07 for the period 1400-1900 CE and 0.14 vs 0.12 for the period 1901-2000 CE). The correlations between our reconstruction and PHYDA are high and spatially extensive for the entire Southern Cone (0.25 and 0.41), except for southern Patagonia. This reflects that both employ the same reconstruction technique, and in part the same
proxy and model data, as PHYDA is based on the CESM model, whose isotope-enabled version is also part of our multi-model ensemble.





**Appendix C: Observation error estimation from the Signal-to-Noise Ratio (SNR)**

We estimate the observation error variance $\mathbf{R}$ (Equation 2) for each proxy record from an assumed signal-to-noise ratio (SNR).

The SNR is defined as the ratio of the standard deviations of the unperturbed time series $T$ and the white noise $N$:

$$\text{SNR} := \frac{\text{std}(T)}{\text{std}(N)} \tag{C1}$$

As the observation error variance $\mathbf{R}$ is by definition equal to the squared standard deviation of the noise $N$, we get for $\mathbf{R}$

$$\mathbf{R} = \text{std}(N)^2 = \text{var}(N) \tag{C2}$$

$$\Rightarrow SNR = \frac{\text{std}(T)}{\sqrt{\mathbf{R}}} \tag{C3}$$

$$\Rightarrow \mathbf{R} = \frac{\text{var}(T)}{\text{SNR}^2} \tag{C4}$$

The variance of $T$ is not known directly because the proxy record time series $Y$ represents the noisy time series $T + N$, whose variance can be used to compute $\text{var}(T)$. When assuming that $N$ and $T$ are uncorrelated, their variances are additive.

$$\text{std}(T+N) = \sqrt{\text{var}(T+N)} \tag{C5}$$

$$= \sqrt{\text{var}(T) + \text{var}(N) + \text{cov}(T,N)} = \sqrt{\text{var}(T) + \text{var}(N)} \tag{C6}$$

$$= \sqrt{\text{var}(T) + \mathbf{R}} \tag{C7}$$

$$\Rightarrow \text{var}(T) = \text{var}(T+N) - \mathbf{R} \tag{C8}$$

Using the relationship from equation C8 in equation C4 we get

$$\mathbf{R} = \frac{\text{var}(T+N) - \mathbf{R}}{SNR^2} \tag{C9}$$

$$\Rightarrow \mathbf{R} = \frac{\text{var}(T+N)}{1 + SNR^2} = \frac{\text{var}(Y)}{1 + SNR^2} \tag{C10}$$

Using equation C10, the observation error $\mathbf{R}$ can be estimated from the variance of the proxy record time series and by
assuming a specific SNR value for the proxy records, which does not need to be the same for all proxy records.



## Appendix D: Proxy record lists

For an overview of all proxy records employed in this regional climate field reconstruction, the proxy records are grouped by climate archive type and presented in tabular form. In the *database* column, the proxy record databases from which the values were taken are named, in case the record is part of one. Additionally the original publications are cited in the *source* column. For the citations of proxy records that were part of a database, the reader is referred to the publications presenting the proxy record databases. The level of detail in the tables varies by climate archive type, as the information from column with unique entries (e.g. the proxy variable, PSM type, Seasonality or SNR) has been transferred to explanatory text above the tables to limit the size of the tables. All data processing steps can be retraced and reproduced with the Jupyter Notebooks accompanying this publication.

### D1    Speleothems

The proxy variables are $\delta^{18}O$ of aragonite and $\delta^{18}O$ of calcite for all processed speleothem time series. For all speleothems, the same *precipitation weighting*-type PSM has been used, thus rendering the application of some additional seasonality restrictions unnecessary. The SNR is computed from the variance of the time series by assuming the same SNR for all speleothems (See Section C). Whereas the speleothem time series have relatively high median temporal resolutions, all time series are resampled to 10 years resolution to conservatively account for smoothing effects in the karst.

For caves with multiple records of similar resolution, composites were computed according to Novello et al. (2021) (step 2 and 3 in section 3.3) by applying the following steps to the time series of various records in one cave during the overlap period 1) resampling all proxy time series to annual resolution, 2) standardizing the time series (zero-mean & standard deviation equal to one) 3) computing the mean of the overlapping values 4) destandardizing the mean time series using the mean and standard deviation of the time series which is longer. Further steps detailed in section 3.3 of Novello et al. (2021) are not applied, as these involve age model ensembles, which have not been used in this study. The practical computation of speleothem composites can be retraced in the Jupyter Notebooks accompanying this publication.

### D2    Lake sediment records

Only lake sediment records directly related to the isotopic composition of precipitation (and thus not mostly affected by additional evaporation) or already calibrated to temperature have been included. To make this distinction, the original publications have been consulted and records have been accordingly selected. This restriction excluded most lake sediment records available in the Iso2k database (Konecky et al., 2020). All lake sediment records are used on a time scale of five years, to account for the effect of bioturbation in the sedimentation process. For all lake sediments, the same SNR value has been assumed, also for temperature calibrated records which come with an error variance.



| Nr | Site name | Location (Lat,Lon) | Time (CE) | Resolution time scale [yrs] | Database (ID in database) | Source | Comment |
|---|---|---|---|---|---|---|---|
| 1 | Dos Anas cave | 22.4,276.0 | 746-2000 | 2 | SISALv3 (443) | Fensterer et al. (2012) | |
| 2 | Tzabnah cave | 20.7,270.3 | 487-2004 | 2 | SISALv3 (147) | Medina-Elizalde et al. (2010) | |
| 3 | Perdida cave | 18.0,293.0 | 1208-2003 | 2 | SISALv3 (378) | Winter et al. (2011) | |
| 4 | Juxtlahuaca cave | 17.4,260.8 | 0-2010 | 2 | SISALv3 (286) | Lachniet et al. (2012) | |
| 5 | Macal Chasm | 16.9,270.9 | 0-1992 | 3 | SISALv3 (178) | Akers et al. (2016) | |
| 6 | Yok Balum cave | 16.2,270.9 | 0-2005 | 1 | SISALv3 (209) | Kennett et al. (2012) | Used longest record from location |
| 7 | Caripe Cave | 10.2,296.4 | 0-1993 | 2 | - | Medina et al. (2023) | |
| 8 | Paraiso cave | -4.1,304.6 | 0-1998 | 7 | SISALv3 (424) | Wang et al. (2017) | |
| 9 | Trapiá cave | -5.6,322.3 | 0-1932 | 3 | - | Utida et al. (2023) | |
| 10 | Shatuca cave | -5.7,282.1 | 0-1984 | 8 | SISALv3 (434) | Bustamante et al. (2016) | |
| 11 | Palestina cave | -5.9,282.6 | 413-1850 | 5 | SISALv3 (94) | Apaéstegui et al. (2014) | |
| 12 | Cascayunga Cave | -6.4,282.9 | 1088-1999 | 1 | - | Bird et al. (2011b) | |
| 13 | Huagapo cave | -11.3,284.2 | 3-1993 | 2 | SISALv3 (597,598) | Kanner et al. (2013) | Composite |
| 14 | Mata Virgem cave | -11.6,312.5 | 166-1814 | 1 | - | Azevedo et al. (2019) | Used Mata Virgem 1 record |
| 15 | Cuíca cave | -11.7,299.4 | 338-2013 | 2 | SISALv3 (752) | Libera et al. (2022) | |
| 16 | Diva cave | -12.4,318.4 | 2-1999 | 4 | SISALv3 (113,146,203) | Novello et al. (2012) | Composite |
| 17 | São Matheus/Bernardo cave | -13.8,313.6 | 264-1998 | 1 | SISALv3 (430,431) | Novello et al. (2018) | Composite |
| 18 | Pau d'Alho cave | -15.2,303.2 | 491-1860 | 1 | SISALv3 (128) | Novello et al. (2016) | |
| 19 | Tamboril cave | -16.0,313.0 | 272-1982 | 2 | SISALv3 (97) | Wortham et al. (2017) | |
| 20 | Umajalanta cave | -18.1,294.2 | 620-1863 | 2 | SISALv3 (499,497,498,518) | Apaéstegui et al. (2018) | Composite |
| 21 | Jaraguá cave | -21.1,303.4 | 422-2000 | 3 | SISALv3 (449) | Novello et al. (2018) | Composite as part of SISAL |
| 22 | Cristal Cave | -24.5,311.4 | 0-2006 | 2 | - | Vuille et al. (2012) | |

**Table D1.** Speleothem proxy records description table

| Nr | Site name | Location (Lat,Lon) | Time (CE) | Resolution | Proxy variable | PSM | Seasonality | Database (ID in database) | Source |
|---|---|---|---|---|---|---|---|---|---|
| 1 | Lago El Grancho | 11.9,274.1 | 341-2004 | 4 | $\delta^{18}$O | prec. weighted | None | Iso2k (263) | Stansell et al. (2013) |
| 2 | Laguna Pumacocha | -10.7,283.9 | 0-2007 | 2 | $\delta^{18}$O | prec. weighted | None | Iso2k (343) | Bird et al. (2011a) |
| 3 | Laguna Chepical | -32.3,289.5 | 0-2005 | 1 | tsurf | season | 11,12,1,2 | Pages2k (SAm_30) | de Jong et al. (2013) |
| 4 | Laguna Aculeo | -33.8,289.1 | 856-1997 | 1 | tsurf | season | 12,1,2 | Pages2k (SAm_3) | von Gunten et al. (2009) |
| 5 | Laguna Escondida | -45.5,288.2 | 400-2008 | 1 | tsurf | direct | None | Pages2k (SAm_31) | Elbert et al. (2013) |
| 6 | Lago Plomo | -47.0,287.1 | 1384-2001 | 1 | tsurf | season | 9,10,11,12,1,2 | Elbert et al. (2015) | Elbert et al. (2015) |

**Table D2.** Lake sediment proxy records description table

1275 ## D3 Sclerosponge

For the conversion of the sclerosponge values from Montego Bay, Jamaica, to temperature, the formula presented in Haase-Schramm et al. (2003) was used. From the two provided records, we selected the record which was located closer the sea surface (see original publication for details). The time series values were resampled to five years to account for the non-annual resolution of the record.



| Nr | Site name | Location (Lat,Lon) | Time (CE) | Resolution | Proxy variable | PSM | Seasonality | SNR | Database (ID in database) | Source |
|----|-----------|--------------------|-----------|------------|----------------|-----|-------------|-----|---------------------------|--------|
| 1 | Montego Bay, Jamaica | 18.5,282.0 | 1356-1991 | 5 | tsurf | direct | None | assumed | Pages2k (150) | Haase-Schramm et al. (2003) |

**Table D3.** Sclerosponge proxy record description table

## D4    Marine sediment

The Cariaco Basin record from Black et al. (2007) was the only marine sediment included in our regional climate field reconstruction, as other marine sediment from South and Central America only provide a longer than decadal resolution. Due to the exceptionally high sedimentation rate in the Cariaco Basin, the record was treated as an annual record according to its temporal resolution.

| Nr | Site name | Location (Lat,Lon) | Time (CE) | Resolution | Proxy variable | PSM | Seasonality | SNR | Database (ID in database) | Source |
|----|-----------|--------------------|-----------|------------|----------------|-----|-------------|-----|---------------------------|--------|
| 1 | Cariaco Basin | 10.8,295.2 | 1221-1990 | 1 | tsurf | season | 3,4,5 | assumed | Pages2k (11) | Black et al. (2007) |

**Table D4.** Marine sediment proxy record description table

## D5    Ice cores

For all ice core $\delta^{18}$O record locations, the precipitation weighting PSM has been applied, rendering the definition of a seasonality unnecessary. For all ice core proxy records, an SNR value was assumed to compute the error variance.

| Nr | Site name | Location (Lat,Lon) | Time (CE) | Proxy variable | Database (ID in database) | Source | Comment |
|----|-----------|--------------------|-----------|----------------|---------------------------|--------|---------|
| 1 | Quelccaya Ice Cap | -13.9,289.2 | 226-2009 | $\delta^{18}$O | Pages2k (SAm_026) | Thompson et al. (2013) | |
| 2 | Illimani | -16.6,292.2 | 1771-1998 | D | Iso2k (485) | Hoffmann et al. (2003) | Deuterium converted to $\delta^{18}$O with factor 1/8 |
| 3 | James Ross Island | -64.2,302.3 | 0-2007 | D | Pages2k (Ant_10) | Abram et al. (2013) | Deuterium converted to $\delta^{18}$O with factor 1/8 |

**Table D5.** Ice core proxy records description table

## D6    Corals

All coral proxy records were used on an annual time scale. For the PSM, the linear PSM for temperature was used, thus all records were calibrated to temperature. Calibration values (including SNR estimates) were calculated for the annual and seasonal reconstruction/calibration separately.



| Nr | Site name | Location (Lat,Lon) | Time (CE) | Resolution | Proxy variable | SNR (Ann.,DJF) | Database (ID in database) | Source |
|---|---|---|---|---|---|---|---|---|
| 1 | Gingerbread Bahamas | 25.8,281.4 | 1552-1991 | 1 | calcification | 0.2, 0.15 | PAGES2K-Ocn_065 | Saenger et al. (2009) |
| 2 | Alinas Reef, Biscayne National Park, Florida | 25.4,279.8 | 1751-1986 | 1 | $\delta^{18}O$ | 0.1,0.13 | Iso2K-255 | Swart et al. (1996) |
| 3 | Dry Tortugas | 24.6,277.7 | 1733-2008 | 1 | Sr_Ca | 0.55,0.97 | PAGES2K-Ocn_070 | DeLong et al. (2014) |
| 4 | Punta Maroma, Mexico | 20.8,273.3 | 1773-2009 | 1 | calcification | 0.88,0.4 | PAGES2K-Ocn_073 | Tierney et al. (2015) |
| 5 | Turrumote Reef, Puerto Rico | 17.9,293.0 | 1751-2004 | 1 | $\delta^{18}O$ | 0.73,0.59 | PAGES2K-Ocn_111 | Kilbourne et al. (2008) |
| 6 | Secas Island, Panama | 8.0,278.0 | 1707-1984 | 1 | $\delta^{18}O$ | 0.27,0.23 | PAGES2K-Ocn_104 | Linsley et al. (1994) |
| 7 | Urvina Bay | -0.4,268.8 | 1607-1981 | 1 | $\delta^{18}O$ | 0.61,0.34 | PAGES2K-Ocn_087 | Dunbar et al. (1994) |

**Table D6.** Coral proxy records description table

## D7 Trees

The employed tree records are described solely in text form rather than tabular format due to their large quantity. The proxy data from trees was mainly taken from three proxy record databases according to the selection criteria outlined in the main text. This selection resulted in 203 tree proxy time series from the South American Drought Atlas (Morales et al., 2020), 42 from Breitenmoser et al. (2014), as used in and published alongside Steiger et al. (2014), and 5 records from the Pages2k database (Emile-Geay et al., 2017). We checked for potential overlaps between these proxy databases and excluded double/triple records. As the SADA database only extends back to 1400CE, six records from it have been replaced by the longer original record data which are available in the NOAA database (see code of this publication for exact documentation). All these data bases use tree ring width as a proxy from trees, and not Maximum Wood Density (MXD). In addition to these tree ring data sources, we used four single records/tree ring composites , namely from the central Altiplano (Morales et al., 2012), the northern altiplano (Morales et al., 2023), the western Amazon (Humanes-Fuente et al., 2020) and near the Perito Moreno glacier in Patagonia (Grießinger et al., 2018). For this last tree proxy record, the proxy variable is $\delta^{18}O$ in wood and not tree ring width, but this record has also been calibrated to instrumental variables as all other tree ring records, because precipitation $\delta^{18}O$ can not be directly related to $\delta^{18}O$ in wood. No seasonality restriction was imposed for using the data as the calibration to instrumental variables for the linear PSM was computed for the annual and summer season separately. All used tree records have annual resolution.