# Peer review of "A continental reconstruction of hydroclimatic variability in South America during the past 2000 years"

_EGUsphere, 2024_

## Referee Comment (RC1)

**Review of Choblet et al. "A continental reconstruction of hydroclimatic variability in South America during the past 2000 years"**

**General Remarks**

This study represents a first attempt to reconstruct regional climate change in South America with a focus on multi-decadal to centennial timescales. It introduces a new approach to reasonably include proxy data of lower than annual resolution by explicitly assimilating the models and observations at different temporal resolutions. As such it is a very welcome and important contribution to the field, as particularly for spatially explicit reconstructions, a multi-frequency method has hitherto not been applied in a real-world continental scale study. I am pleased to see that that earlier attempts to reconstruct climate fields in South America are now being improved in terms of methods, data handling and spatial coverage.

Naturally, such a reconstruction comes with the substantial challenge to verify its results, as no instrumental data are available to test the performance on multidecadal and lower timescales. The authors have addressed this issue by cross validation with individual records and several sensitivity tests.

While I think from a methodological perspective this study deserves publication (although I am not a DA expert), I am not fully convinced by the quality of the reconstruction product and thus the climatological interpretations.

The reconstruction shows little to no skill in the internal validation in the key (SASM) area. The same is true for the instrumental validation (Section B2). The latter is not surprising as the signal at higher than multidecadal frequencies comes from remote proxies. However, it worries me that even the longer-term fluctuations in the instrumental data Fig. AF B3 are entirely missed by the reconstruction. Unfortunately, I cannot provide a clear recipe on how to improve this situation and I acknowledge the amount of work the authors have invested to assess the robustness of the results. Maybe more clearly flagging this limitation throughout the paper already helps a lot. In the following I provide some further suggestions. I do not expect the authors to do all of this, it should rather be seen as a collection of options. I am sure the authors have a better understanding than myself about which are the most reasonable things to do.

- The reconstruction seems to perform best outside of the range of the speleothem data in southern South America, where other reconstructions already exist. Maybe it is worth checking if the multi timescale approach improves the reconstructions in this area by comparing the high-resolution (trees) vs. all-proxies reconstructions (there are a few low-resolution data from this area) with the multidecadal fluctuations in the instrumental records and the SADA?
Similar in the NW-corner of the area in Central America, which seems to be the only region with reasonable COE values. I understand this is not the region of interest, but to validate the method it may still be useful?
- The key problem seems to be that the speleothem records disagree quite strongly (as suggested by Figs. 2 and AF19). Are they out of phase due to dating problems or weak climate signals? Or is it a true signal with small scale variations in hydroclimate, which the models fail to reproduce? Given the lack of skillful validation it seems necessary to elaborate more on this. Partly this is done in the manuscript e.g. on page 13, but for example it may be clarified if the mentioned dipole is entirely missing in the model data

or show some more local time series from the recons to identify potential issues. Some use of and reference to the nice video material may also help.
Or: which frequencies are responsible for the low COEs? Fig .5 suggests that the signal is coming from the speleothems at frequencies below 150 years. If the problem is mainly at the higher frequencies, other records may be responsible.

- It is mentioned several times that the reconstruction compares well with local proxy record studies (even in the abstract), but this is not illustrated in the paper. Maybe a more explicit comparison including illustrations would help understanding how the reconstruction incorporated and weighted the low-frequency information from these records? This may also help to identify the most reliable bits of the reconstruction in time and space.

- I am obviously biased here, but I miss the inclusion of the documentary records in this study. South America has the best collection of hydroclimate-related records in the Southern Hemisphere and it is almost a shame not to use this beautiful dataset. They could be included as proxies as in the Neukom et al. reconstructions, but they may also be used for further validation. They are clearly high-resolution datasets so may be of limited use here, but some records include multi-year droughts or pluvials, which may be compared with the results. At least it should be stated why these records are excluded, given that they are even mentioned in the introduction. Some other records from earlier collections are also missing and it is not clear why (see below).

In summary, a revision of the paper may benefit from a more careful treatment of the largely unknown reliability of the reconstruction in the monsoon area, be it by changes in focus and/or wording, a more explicit treatment of the issue as suggested, or otherwise.

**Specific / minor suggestions**

Line 13-14: I don't understand the part of the sentence starting with "with exceptions in ..". Are the regions an exception to the non-uniform pattern? How? I also did not find this statement to be described in detail in the main text.

L. 14: As stated above it is not really shown how the reconstructions align with local studies. So this should either be included or the statement removed.

L. 16-17: To be precise, Neukom et al. 2014 have used a speleothem record in their field reconstruction. This is picky, but I anyways think the key advancement here is that the lower resolution records can now be adequately treated in the multi-timescale approach in contrast to the very simple way they have been used earlier. So this is what should be emphasized in this sentence in my opinion.

L. 24: I am not sure about the "relatively stable external forcing", considering the volcanic eruptions and their large influence in triggering cold periods. Maybe rephrase.

L. 37: "Widely unknown" is a pretty strong statement. I do not disagree, but at least it should be recognized that there is existing work on this. Currently the statement reads as if there wasn't. Please change or include some reference to acknowledge this, it can also be the regional chapter from the latest IPCC report.

L 43ff: I think it is worth including the key review paper by Prieto and García Herrera here (doi: 10.1016/j.palaeo.2008.07.026)

L 51-52: Could cite Neukom & Gergis 2012 (Southern Hemisphere proxy data review) here (doi: 10.1177/0959683611427335).

L58: Also include Boucher et al. 2011 (doi: 10.5194/cp-7-957-2011) here.

L60ff: I missed the Vuille et al. (2012) reference in the section about speleothems (although I saw it was used later on in the manuscript).

L93ff: This section contains quite some details about the methods and should largely be moved into the methods section. A brief description of PaleoDA and what it does and where it was used is sufficient here.

Section 2.1: As mentioned above there are some records that were used in earlier collections or hydroclimate reconstructions (Boucher et al. 2011, Neukom and Gergis 2011, Neukom et al. 2014) that were excluded, and it seems to me some of them would fulfill the criteria mentioned on page 6.:

Documentary records (https://www.ncei.noaa.gov/access/paleo-search/study/8703)

Lake Sediments: Puyehue (Boës and Fagel (2008) used in Boucher11 & Neukom14), Potrok Aike (Haberzettl et al. (2005) used in Boucher11), El Junco (Conroy et al. (2009) used in Neukom14)

Marine Sediment 106KL East Pacific-Peru (Rein (2007), used in Boucher11 & Neukom14)

L228: Maybe clarify, why a selection of 100 years was used and not the full collection. I understand it is due to computational limitations, but this may be stated here.

L242ff: Move this paragraph describing the history of DA up to the beginning of the methods description.

L254ff: The procedure to combine the low- and high-resolution data is not so easy to understand. An illustration similar to Fig. AF2 may be quite helpful.
Also, as far as I understand, the 10-year blocks are non-moving. Please clarify in line 256.

L297ff (C): There are PSMs for tree rings and coral existing. Please clarify why statistical calibration was preferred here.

L318: For better understanding I would move the statement of what values were used for the SNR up here right after the Smerdon citation.

L358ff: Validation: The COE is based on a validation period, and I do not understand, which periods were selected here. The full period that each withheld dataset covers? Please clarify. This may also be helpful to understand why the values are so low.
It is good to see that the DA improves the values but in fact as long as they remain below zero the reconstruction performs worse than a flat line, which seems to be the case over the key area of interest. In the text, this may be misunderstood.
Is it possible to show the development over time? Potentially this would help identifying periods, where the reconstruction is more reliable, because the proxies show better agreement?

L440: I find this interesting. Is it a correct interpretation that the models with the largest variations tend to also lead to the reconstructions with the largest variability? This may be clarified.
In general with DA it is a clear sign of a noisy reconstruction if the results just show small

variability (at least in my experience). This may also be stated somewhere, for example to explain why the tree-only (and partly No-speleo) and LMR hydroclimate recons are basically a flat line in many instances...

L450: I think it should be AF11 not AF12.

---

## Author Response (AR1)

**Summary of changes for revised manuscript: A continental reconstruction of hydroclimatic variability in South America during the past 2000 years (egusphere-2024-545)**

Mathurin A. Choblet, Janica C. Bühler, Valdir F. Novello, Nathan J. Steiger, and Kira Rehfeld

July 20, 2024

**Summary of changes**

Dear Kathleen Wendt,

Thank you for your positive assessment of the proposed modifications.

In response to the helpful suggestions by the two reviewers, we implemented the following changes:

- We have extended the discussion of the validation of the reconstruction, both for the internal and instrumental validation, which clarifies the skill and limitations of our reconstruction.

- We now include eleven historical proxy records and one annual lake record in the reconstruction via linear Proxy System Models as for the tree rings. The proxy record tables in the appendix have been adjusted accordingly.

- Due to the inclusion of new proxy records in the reconstructions, most figures have been updated, as well as the supplement videos, which are now also mentioned in the main text. The code on Github and data on Zenodo have been updated accordingly.

- We reordered and reorganized the appendix and the supplement figures in order of appearance in the main manuscript.

- We included Figure AF12a into the main manuscript (Figure 5, respectively here Figure 6) and removed Figure AF5-AF9 altogether to facilitate the lecture of the manuscript. Figures to illustrate the multi-time scale algorithm have been added (AF B2,AF B3, respectively here Figure 4 and Figure 5).

- We revised the text throughout the manuscript to clarify statements.

- We checked the figure colorschemes for best accessibility.

A detailed response to the helpful remarks of the referees, is given below. The original report is cited in italics, our reply is written in blue color. Sections describing specific adjustments that we planed to do are marked in bold, and we mark in green if we have performed the changes we suggested and partially make additional comments. Line numbers in the green sections refer to the line numbers in the revised manuscript.

**1 Reply to Raphael Neukom**

*This study represents a first attempt to reconstruct regional climate change in South America with a focus on multi-decadal to centennial timescales. It introduces a new approach to reasonably include proxy data of lower than annual resolution by explicitly assimilating the models and observations at different temporal resolutions. As such it is a very welcome and important contribution to the field, as particularly for spatially explicit reconstructions, a multi-frequency method has hitherto not been applied in a real-world continental scale study. I am pleased to see that that earlier attempts to reconstruct climate fields in South America are now being improved in terms of methods, data handling and spatial coverage.* We thank the reviewer for this positive assessment.

*Naturally, such a reconstruction comes with the substantial challenge to verify its results, as no instrumental data are available to test the performance on multidecadal and lower timescales. The authors have addressed this issue by cross validation with individual records and several sensitivity tests.*

*While I think from a methodological perspective this study deserves publication (although I am not a DA expert), I am not fully convinced by the quality of the reconstruction product and thus the climatological interpretations.*

*The reconstruction shows little to no skill in the internal validation in the key (SASM) area. The same is true for the instrumental validation (Section B2). The latter is not surprising as the signal at higher than multidecadal frequencies comes from remote proxies. However, it worries me that even the longer-term fluctuations in the instrumental data Fig. AF B3 are entirely missed by the reconstruction. Unfortunately, I cannot provide a clear recipe on how to improve this situation and I acknowledge the amount of work the authors have invested to assess the robustness of the results. Maybe more clearly flagging this limitation throughout the paper already helps a lot. In the following I provide some further suggestions. I do not expect the authors to do all of this, it should rather be seen as a collection of options. I am sure the authors have a better understanding than myself about which are the most reasonable things to do.*

We thank the reviewer for the critical assessment of our validation. Our study represents the first attempt to systematically include speleothems into PaleoDA reconstructions via a

multi-time scale approach. The validation tools we have employed represent the panoply of usually employed validation tools, and as we outline in the following, the resulting metrics give scores comparable to existing reconstructions. **We think that further clarifications and explanations will help emphasize the skill of our reconstruction and at the same time improve stressing its limitations, in order to motivate more conclusive future validations of multi-timescale PaleoDA reconstructions.** The inclusion of speleothems into climate field reconstructions is an active field of research, and we can also confidently tell, that yet to be published work based on other Data Assimilation techniques yield similar results for central south America (private communication, Lyu, 2023).

Action: Done. See comments below.

Regarding the doubts for the internal validation: **we use the entire reconstruction period for the computation of skill scores**, which differs from other climate field reconstruction techniques. In PaleoDA, the left-out proxy records are usually omitted for the entire reconstruction period, and there is no direct way to estimate the skill over time. **One key aspect of our reconstruction is the comparison to the prior, which can be more meaningful than the actual skill scores. As pointed out by Cook et al. (1999) and Hakim et al. (2016), the coefficient of efficiency (COE) can misleadingly yield negative skill scores due to proxy records and reconstructions having different mean values. Comparing the COE to the prior shows that our reconstruction has skill, also in the core monsoon region, with values comparable or better than in Hakim et al. (2016) and Tardif et al. (2019). While these are global studies, the comparison demonstrates that our values are acceptable for a PaleoDA reconstruction. We will update Figure 2 to only include the period covered by climate models (850-1850 CE), leading to minor changes. One issue related to the apparently low skill scores is the aspect of spatial smoothing. High spatial variability in proxy records can lead to low skill scores in internal validation, which we will add in the discussion of the internal validation. We acknowledge that internal validation in PaleoDA still needs refinement and establishment of clear protocols.**

Action: Done. The additions to the text outlined above have all been included in the manuscript in the corresponding sections, Internal validation (line 386ff, line 410ff) and Conclusion (line 685ff).

Additionally, we thank the reviewer for the suggestions regarding instrumental validation. The grid cell-based skill scores indicate skill in the core monsoon region (AF B1) in terms of correlation and CRPSS, even though high interannual fluctuations are missed. The key limitation is the lack of high-resolution proxies in the core monsoon region: annual

tree rings are remotely located, and speleothems are only used at decadal resolution. With at most 5 data points per speleothem in the instrumentally more reliable second half of the 20th century, meaningful validation is not possible. The shortness of the instrumental record also prevents validating the magnitude of reconstructed centennial hydroclimate changes (Figure 4 in manuscript), which as discussed is also prior model and season dependent, but are clearly visible and persist during MCA and LIA. We will mention the uncertainty of the magnitudes in the discussion.

Our inclusion of speleothems relies on the assumption that their $\delta^{18}$O signatures capture monsoon variability, locally validated in Moquet et al. (2016) and Jiménez-Iñiguez et al. (2022). Model prior correlation fields (AF20 in manuscript) support, that this assumption is conveyed in our reconstruction. **We will highlight this assumption and its need for further validation.** In future work, higher-resolution speleothem data could improve reconstructions if they effectively capture SASM fluctuations on shorter than decadal time scales. An important caveat for instrumental validation is the inconsistency between precipitation datasets, particularly pronounced in the core monsoon region (Figures 1, 2) compared to the rest of the continent. **This complicates validation and will be discussed in the appendix and conclusion.**

Action: Done. See the modifications in the Validation Section (line 372ff), in the Conclusion (line 681ff) and Appendix B2 for emphasizing this key limitation of our study.

In summary, our modifications regarding the validation will:

- explain the importance of comparing validation skills with the prior, update Figure 2 for 850-1850 CE.
- Mention high spatial variability in proxy records leading to low internal validation scores.
- Discuss difficulties in instrumental validation, particularly for decadal-centennial scales.
- Highlight the speleothem $\delta^{18}$O-precipitation assumption and need for future validation.
- Emphasize inconsistencies in instrumental precipitation datasets complicating validation.

Action: Done.

[Figure]

Figure 1: Correlation of annual mean precipitation values in the CRU (Harris et al., 2020), GPCC (Schneider et al., 2008) and 20CR (Compo et al., 2011) datasets. The correlation is computed separately for the period 1900-1950 CE, prior to the onset of radiosonde precipitation monitoring (1958), and for the second half of the 20th century. The inset box computes the average correlation for all grid cells (all), for the core monsoon region (mon.) and for all grid cells except the core monsoon region (rest).

[Figure]

Figure 2: Annual core monsoon region precipitation anomaly from CRU (Harris et al., 2020), GPCC (Schneider et al., 2008) and 20CR (Compo et al., 2011)

- *The reconstruction seems to perform best outside of the range of the speleothem data in southern South America, where other reconstructions already exist. Maybe it is worth checking if the multi timescale approach improves the reconstructions in this area by comparing the high-resolution (trees) vs. all-proxies reconstructions (there are a few low- resolution data from this area) with the multidecadal fluctuations in the instrumental records and the SADA? Similar in the NW-corner of the area in Central America, which seems to be the only region with reasonable COE values. I understand this is not the region of interest, but to validate the method it may still be useful?*

From the improved metric described in the comments above, it is not apparent, that the Southern part of South America is better reconstructed than the rest. Performing the suggested analysis, we compute the tree-ring-only reconstruction (Figure 3a) and compare it to the results to using all proxy records (Figure 3b). The correlation for the tree-ring-only reconstruction improves slightly yielding 0.15 instead of 0.12 for the period from 1400-1900 CE, whereas the score in the 20th century slightly decreases from 0.13 to 0.12. The comparison is complicated by two factors. First, the tree-ring-only reconstruction uses a dataset closer to the one from the SADA and, thus, may automatically yield slightly higher results. Second, we are computing the SPEI and not the scPDSI metric, which adds a caveat to the comparison. We opt to leave a proper comparison between the PaleoDA drought reconstruction and its assessment on various time scales to future studies. This type of assessment would be particularly interesting when both tree-rings and speleothems are available for a region, as they exhibit sensitivity to climate variability on different time scales, but in our current proxy record dataset the regions covered by these proxy records are quite distinct.

Action: As explained, no changes have been performed here.

- *The key problem seems to be that the speleothem records disagree quite strongly (as suggested by Figs. 2 and AF19). Are they out of phase due to dating problems or weak climate signals? Or is it a true signal with small scale variations in hydroclimate, which the models fail to reproduce? Given the lack of skillful validation it seems necessary to elaborate more on this. Partly this is done in the manuscript e.g. on page 13, but for example it may be clarified if the mentioned dipole is entirely missing in the model data or show some more local time series from the recons to identify potential issues. Some use of and reference to the nice video material may also help. Or: which frequencies are responsible for the low COEs? Fig .5 suggests that the signal is coming from the speleothems at frequencies below 150 years. If the problem is mainly at the higher frequencies, other records may be responsible.*

As discussed above, the reconstruction probably contains more skill than indicated by

**Tree ring only reconstruction**

[Figure]

(a) Figure AF B4 in the original manuscript comparing scPDSI from the SADA (Morales et al., 2020) and PHYDA (Steiger et al., 2018) using only tree rings as proxy records in our reconstructions

**All proxy reconstruction**

[Figure]

(b) Figure AF B4 in the original manuscript comparing scPDSI from the SADA (Morales et al., 2020) and PHYDA (Steiger et al., 2018) using all proxy records

Figure 3

the raw CE values. At the decadal to centennial time scales we are examining, dating uncertainties of speleothem proxy records should not play a significant role. Therefore, we generally assume that the speleothem $\delta^{18}$O signatures represent real hydroclimatic variations. As discussed in the discussion section of our study, we suggest that the PaleoDA reconstruction method smooths out climate fields and is, thus, not able to represent small spatial variability, which indeed is a limitation of the reconstruction. As such, we will clarify line 495 as follows:

"In addition, looking into the spatial correlations of the mean $\delta^{18}$O of precipitation values for that region in the model simulations **only shows a dipole in two out of the five climate model** simulations (AF18) and thus reveals the benefit of combining proxy and model data with PaleoDA. **However, more research for quantifying the extent and possible spatio-temporal variations of the South American $\delta^{18}$O and precipitation dipole is required, also by incorporating additional proxy records from NEB as we currently only employ two.**"

Action: Done.

- *It is mentioned several times that the reconstruction compares well with local proxy record studies (even in the abstract), but this is not illustrated in the paper. Maybe a more explicit comparison including illustrations would help understanding how the reconstruction incorporated and weighted the low-frequency information from these records? This may also help to identify the most reliable bits of the reconstruction in time and space.*

Thank you for pointing out this inconsistency. We realize that the employed wording can suggest that one-to-comparison to other studies have been performed, in particular in the abstract, which we have not done. We discuss similarities in trends for South American climate field reconstruction at the beginning of the discussion of our study by comparing the main conclusions about the most important climate changes during the past 2000 years from the named studies and concluding that our climate trends match the ones previously described. A one-to-one comparison between our quantitative climate field reconstruction and proxy-record interpretations from single or multi-proxy studies are not straight forward to do, and we would not consider them particularly useful, as we have employed the same proxy records (and more) as used in the named studies. Furthermore, $\delta^{18}$O is a proxy for large scale regional atmospherical behaviour, and does not necessarily match local hydrological conditions. Agreement with local hydrology (or records) is not expected. **Therefore, we will revise the abstract and main text to make this clear. In particular, we will replace "compare/comparison" accordingly.** The idea to quantify how the reconstruction weighted the different time scales is indeed interesting and

important. While we restrict ourselves to comparing reconstructions with and without speleothems, we conclude that these carry important information on centennial climate change. A more quantitative assessment of lower frequencies and weighting of different timescales will be left to future studies.

Action: Done. See the changes in the Introduction (line 120) and Section 6.2 (line 558ff).

- *I am obviously biased here, but I miss the inclusion of the documentary records in this study. South America has the best collection of hydroclimate-related records in the Southern Hemisphere and it is almost a shame not to use this beautiful dataset. They could be included as proxies as in the Neukom et al. reconstructions, but they may also be used for further validation. They are clearly high-resolution datasets so may be of limited use here, but some records include multi-year droughts or pluvials, which may be compared with the results. At least it should be stated why these records are excluded, given that they are even mentioned in the introduction. Some other records from earlier collections are also missing and it is not clear why (see below).*

Thank you for the suggestion. We agree that it should be clearly stated why we are not including historical proxy records. At the moment of the submission of this manuscript, we were not aware of any systematic inclusions of historical precipitation indices into past PaleoDA reconstructions. Not being experts on these type of proxy data, we considered their inclusion into PaleoDA methodologically unclear regarding the forward model due to historical indices covering only a small range of discrete values (-2,-1,0,1,2). However, the situation seems to have changed recently, as a recent global PaleoDA reconstruction by Valler et al. (2024) includes the historical data collection by Burgdorf et al. (2023), which also contains the Neukom et al. (2009) data. Valler et al. (2024) included historical indices via a linear regression forward model. Therefore, we will procede similarly and include historical indices from Neukom et al. (2009) into a revised reconstruction using a linear forward model to temperature or precipitation, based on the correlation and p-value scores. However, we do not expect our main findings of the study regarding centennial climate changes in South America to be affected by including these records due to temporal limitations. In particular, we compare the results with the South American Drought Atlas to see if a positive impact of the historical records can be found.

Action: Done. The historical documentary indices are now included in our reconstruction (see Figure 1 and Appendix Table 7). Calibrated to instrumental temperature, SPEI and precipitation (as for the tree ring records), they exhibit SNR values from 0.21 to 0.74. No noticeable impact on the validation of the South American Drought Atlas could be found (Appendix C3). The internal validation skill scores (Section 4) have decreased in

terms of correlation, but increased in terms of the Coefficient of Efficiency.

In the detailed comments, further lake and marine sediments are suggested. We established the second inclusion criterion, (line 140ff), that demands that in the original publication of the proxy record a linear relationship between the proxy variable and environmental variables is established (e.g. via a linear regression or another forward model). We assessed that this is not the case for 106KL East Pacific-Peru (Rein, 2007) and Potrok Aike (Haberzettl et al., 2005) records. For the El Junco Lake record (Conroy et al., 2008), it seems as the part of the proxy record time series published alongside Neukom et al. (2014) has been linearly interpolated to an annual resolution and is not actually annual. For the linear regression based forward models, we require annual resolution and can, therefore, not include this record. For the Lago Puyehue record (Boës and Fagel, 2007), indeed all our proxy inclusion criteria are met and we will, therefore, include it in our revised reconstruction with a calibration to precipitation as performed in the original study. As this part of the Andes is densely covered tree-ring and lake-record wise, we do not expect this record to have a large impact on our reconstruction, but we are keen on including it to be as complete as possible. However, we also point out, that there might be further records that we have missed, although it might have met out initial selection criteria. Our current selection of records is still substantial and, thus, leads to a solid analysis.

Action: Done. See the adjusted lake proxy record table (Appendix Table 2).

*In summary, a revision of the paper may benefit from a more careful treatment of the largely unknown reliability of the reconstruction in the monsoon area, be it by changes in focus and/or wording, a more explicit treatment of the issue as suggested, or otherwise.*

The issues will be adressed via a more precised description and explanation of the validation results. We will also highlight the difficulties of validating the reconstruction in the conclusion of the study.

Action: Done. See changes in the Appendices C1 and C2 and in the Conclusion (line 681ff).

**1.1 Detailed Comments**

*Line 13-14: I don't understand the part of the sentence starting with "with exceptions in ..". Are the regions an exception to the non-uniform pattern? How? I also did not find this statement to be described in detail in the main text.*
Thank you, for pointing out the confusing sentence structure. It is meant that climate changes in northeastern Brazil and the Southern Cone are not in phase with the rest of

the continent. We will change the sentence as follows:

".. . The reconstruction reveals anomalous climate periods: a wetter and colder phase during the Little Ice Age ($\sim$1500 - 1850 CE) and a drier, warmer period corresponding to the early Medieval Climate Anomaly ($\sim$600 - 900 CE). However, these patterns are not uniform across the continent, **with climate trends in northeastern Brazil and the Southern Cone not following the patterns of the rest of the continent**, indicating regional variability."

The according part in the main text starts in line 398:

" . . . The reconstructed patterns are mostly homogeneous over the continent and show a trend towards colder and wetter conditions during the LIA, especially for the central and northern part of the continent. The Southern Cone, however, experienced warmer and drier conditions during the LIA. Warmer and drier conditions are predominant during the transition period, in particular preceding the MCA, except for the Southern Cone and the Nordeste (North Eastern Brazil)."

Action: Done.

*L. 14: As stated above it is not really shown how the reconstructions align with local studies. So this should either be included or the statement removed.*

As stated in the reply above, we will clarify our wording regarding the qualitative comparison to insights from local speleothem studies and why one-to-one comparisons are not performed.

*L. 16-17: To be precise, Neukom et al. 2014 have used a speleothem record in their field reconstruction. This is picky, but I anyways think the key advancement here is that the lower resolution records can now be adequately treated in the multi-timescale approach in contrast to the very simple way they have been used earlier. So this is what should be emphasized in this sentence in my opinion.* Thank you for extracting our key advancement more clearly than we did. We will change the section as follows:
"...Despite methodological uncertainties regarding climate model biases and proxy record interpretations, this study marks a crucial first step in incorporating **low resolution proxy records as speleothems** into climate field reconstructions **using a multi-timescale approach. Adequately extracting and using the information from speleothems** potentially **enhances** insights into past hydroclimatic variability and hydroclimate projections...."

Action: Done.

*L. 24: I am not sure about the "relatively stable external forcing", considering the volcanic eruptions and their large influence in triggering cold periods. Maybe rephrase.*
Thank you. We will change the section as follows:

"It provides rather stable**, close to present-day boundary** conditions prior to the onset of industrialization with relatively constant **greenhouse gas concentration and sea level, and climate variability due to natural, solar and volcanic forcings. Thus, it** represents a well-studied benchmark for climate models (e.g. Jungclaus et al., 2017)."

Action: Done.

*L. 37: "Widely unknown" is a pretty strong statement. I do not disagree, but at least it should be recognized that there is existing work on this. Currently the statement reads as if there wasn't. Please change or include some reference to acknowledge this, it can also be the regional chapter from the latest IPCC report.*
Thank you for pointing this out. We will change the sentence accordingly:

"However, the broader effects of anthropogenic climate change on the entire hydrological cycle and its variability, including changes in precipitation extremes and the occurrence of droughts, remain less studied than for temperature."

*L 43ff: I think it is worth including the key review paper by Prieto and García Herrera here (doi: 10.1016/j.palaeo.2008.07.026)*
*L 51-52: Could cite Neukom & Gergis 2012 (Southern Hemisphere proxy data review) here (doi: 10.1177/0959683611427335).*
*L58: Also include Boucher et al. 2011 (doi: 10.5194/cp-7-957-2011) here.*
*L60ff: I missed the Vuille et al. (2012) reference in the section about speleothems (although I saw it was used later on in the manuscript).*
Thank you for providing all these helpful references which we will thankfully include at the respective sections.

Action: Done. See modifications in the Introduction.

*L93ff: This section contains quite some details about the methods and should largely be moved into the methods section. A brief description of PaleoDA and what it does and where it was used is sufficient here.*

In this section, we introduce PaleoDA with its benefits and limitations and briefly explain how we plan to overcome certain limitations. The detailed description is followed

in the methods section. We consider it important to stress the methodology in the introduction already, as it also highlights the methodological nature of our study and PaleoDA provides an easy framework for the inclusion of speleothems into climate field reconstructions. Furthermore, the five isotope-enabled models have not been employed jointly for climate field reconstructions. To stress the methodological novelty of our approach, we will adjust this part as follows:

" ...PaleoDA does not directly rely on gridded instrumental datasets. The climate simulations provide time series long enough to also include proxy records of relatively low resolution, such as speleothems, which cannot be calibrated to instrumental data. We use five state-of-the-art isotope-enabled climate simulations, which simulate the isotopic composition of precipitation and which were made publicly available recently (Bühler et al., 2022). **Employing these previously unused simulations in PaleoDA provides a twofold information gain:** first, it enables the inclusion of speleothem records in the PaleoDA without the uncertainties associated with instrumental calibration, and second, it facilitates the comparison of multiple simulations of rainfall $\delta^{18}$O values from different models, thereby mitigating biases stemming from individual proxies and models."

Action: Done.

*Section 2.1: As mentioned above there are some records that were used in earlier collections or hydroclimate reconstructions (Boucher et al. 2011, Neukom and Gergis 2011, Neukom et al. 2014) that were excluded, and it seems to me some of them would fulfill the criteria mentioned on page 6.: Documentary records Lake Sediments: Puyehue (Boës and Fagel (2008) used in Boucher11 & Neukom14), Potrok Aike (Haberzettl et al. (2005) used in Boucher11), El Junco (Conroy et al. (2009) used in Neukom14) Marine Sediment 106KL East Pacific-Peru (Rein (2007), used in Boucher11 & Neukom14)* This is answered in the major comments.

*L228: Maybe clarify, why a selection of 100 years was used and not the full collection. I understand it is due to computational limitations, but this may be stated here.*
Computational limitations are one issue, but the main reason is that pseudoproxy experiments (performed by us but also found in previous PaleoDA studies) suggest improved reconstruction metrics when only using a subset of the full ensemble for the covariance matrix and repeating the reconstructions with the Monte Carlo procedure. We will clarify this in section 3.5.

A reference to section 3.5 will be added in line 226: "The prior covariance matrix $\hat{X}$ is used from already computed simulations instead of restarting the model ensemble and thus represents the climatological covariance. To compute it, we use an ensemble of 100 randomly selected simulation years (**see Section 3.5**)".

Line 349 in section 3.5 will be adjusted as follows:
"We use a Monte Carlo technique of repeating the reconstructions 50 times with different ensembles of 100 randomly selected simulation years and using 80% of all proxy records in each repetition similar to Hakim et al. (2016) and Tardif et al. (2019). Doing so improves the representation of the reconstruction uncertainty, attenuates the effect of outliers in the proxy record selection **and the prior ensemble provided by the climate model simulations as suggested by Pseudoproxy Experiments**. ..."
Action: Done.

*L242ff: Move this paragraph describing the history of DA up to the beginning of the methods description.*
Thank you for pointing this out. We will move the section to L208ff.

Action: Done.

*L254ff: The procedure to combine the low- and high-resolution data is not so easy to understand. An illustration similar to Fig. AF2 may be quite helpful. Also, as far as I understand, the 10-year blocks are non-moving. Please clarify in line 256.*

Thank you for the suggestions. We will add an Appendix Figure for sketching how we employ the multi-time step approach.
Indeed, the 10-year blocks of the prior are non-moving (However, in each Monte Carlo iteration of the reconstruction, a new ensemble of 100 randomly selected prior ensemble members is constructed (see comment above)). The fact that the 10-year blocks are non-moving will be clarified as follows in line 256:
"Instead of reconstructing the target time period (1-2000 CE) year by year, we divide it into decadal blocks, as we will use the speleothem values on a decadal time scale only. Additionally, we reconstruct annual and quinquennial time scales as sub-blocks of the decadal block. **The decadal prior block is the same during the entire reconstruction period.** ..."

Action: Done. See the additional illustrations and their captions in Figure 4 and 5.

*L297ff (C): There are PSMs for tree rings and coral existing. Please clarify why statistical calibration was preferred here.*

Thank you for the comment. We will clarify more specifically why the statistical PSM was chosen:

"For proxy records that can be calibrated to instrumental data, such as corals and tree

$$\mathbf{X}^f = \begin{pmatrix} \mathbf{X}_{120} \\ \mathbf{X}_{50} \\ \dots \\ \mathbf{X}_{930} \end{pmatrix} \Rightarrow \overbrace{\begin{pmatrix} \mathbf{X}_{120} & \mathbf{X}_{121} & \dots & \mathbf{X}_{129} \\ \mathbf{X}_{50} & \mathbf{X}_{51} & \dots & \mathbf{X}_{59} \\ \dots & \dots & \dots & \dots \\ \mathbf{X}_{930} & \mathbf{X}_{931} & \dots & \mathbf{X}_{939} \end{pmatrix}}^{\text{block size=10 years}} \Big\} \; N_y \text{ ensemble members}$$

Figure 4: Illustration of how single-time scale PaleoDA prior, consisting of a random collection of climate fields, is extended into a matrix which also contains the subsequent years for multi-time scale PaleoDA. Each $\mathbf{X}_i$ corresponds to the mean climate field of one simulation year, the index denotes the year of the simulation that was randomly selected for creating the ensemble. The $\mathbf{X}_i$ could also be depicted explicitly as a vector, thus rendering the matrix three-dimensional. In the multi-time scale PaleoDA, rows of the matrix are averaged over several years in order to assimilate multiyear means.

[Figure]

Figure 5: Assignment of non-annually resolved proxy records to 5 and 10 year time scales (e.g. lake records and speleothems in our reconstruction). During the multi-time scale PaleoDA, the values of these types of proxies are assigned to the 5 or 10 year block means (see Figure 4 instead of annual values. To facilitate the assignment to the blocks, the non-annually resolved proxy records are resampled to 5 and 10 year resolutions as described in the main text.

rings, we employ a linear regression- based PSM **as usually employed in PaleoDA** (e.g. Hakim et al., 2016; Steiger et al., 2018; Tardif et al., 2019, **King et al. (2021),Sanchez et al. (2021)**). **While more specific PSMs for corals and tree rings exist, the linear-regression based PSMs also yield similar reconstruction results in PaleoDA (Dee et al., 2016). More complex PSMs for corals and tree rings also require environmental variables for sea water and air moisture, which are not available for all employed climate model simulations. Furthermore, we consider it preferable to use this type of univariate linear PSMs in PaleoDA, as the covariance relationship between observations and reconstructed climate field remains more tractable.** Linear regression equations . . ."

    Action: Done.

    *L318: For better understanding I would move the statement of what values were used for the SNR up here right after the Smerdon citation.* Line 327 will be moved up to line 318, such that the text will read as follows:

" For all other records in the non-instrumental era, we express the proxy record error in terms of the signal-to-noise ratio (SNR), assuming Gaussian and timescale-independent noise (Smerdon, 2012). **We assume an SNR of 0.5 for all proxy records which we have not calibrated to instrumental variables, following exploratory studies by Wang et al. (2014) and Orrison et al. (2022).** The SNR can then be converted into the entries of the observation error matrix $R$ . . ."

    Action: Done.

    *L358ff: Validation: The COE is based on a validation period, and I do not understand, which periods were selected here. The full period that each withheld dataset covers? Please clarify. This may also be helpful to understand why the values are so low. It is good to see that the DA improves the values but in fact as long as they remain below zero the reconstruction performs worse than a flat line, which seems to be the case over the key area of interest. In the text, this may be misunderstood. Is it possible to show the development over time? Potentially this would help identifying periods, where the reconstruction is more reliable, because the proxies show better agreement?*

    As outlined in the response to the major comments, in PaleoDA the COE is usually computed for the entire reconstruction period, in contrast to methods based on the regression of principal component regression which allow for a straight forward way to compute the evolution of the COE over time. See above for more comments on this important issue and why the obtained results are comparable to other PaleoDA reconstructions.

    *L440: I find this interesting. Is it a correct interpretation that the models with the*

*largest variations tend to also lead to the reconstructions with the largest variability? This may be clarified. In general with DA it is a clear sign of a noisy reconstruction if the results just show small variability (at least in my experience). This may also be stated somewhere, for example to explain why the tree-only (and partly No-speleo) and LMR hydroclimate recons are basically a flat line in many instances...*

From the equations of the EnKF it can not be stated that models with the largest variations also automatically result in the reconstructions with the largest variability. While larger variations might possibly lead to larger covariances, which are the mechanism behind the EnKF, the covariances are also partly normalised. Indeed, the prior dependency of the reconstruction is an important aspect of our results and we are considering highlighting it more in the results and main section by including AF12a into the main text. We will also highlight the difference in the correlation fields of the different models visible in AF18 and AF20, which hints at how proxies from different regions have a varying influence on the reconstruction depending on the prior. For instance AF20 shows that the SASM precipitation index in the models is correlated to $\delta^{18}O$ in NEB Brazil only in some models.

The topic warrants further quantitative investigation, building on the already mentioned studies by Sanchez et al. (2021) and Parsons et al. (2021) who mentioned model covariance/teleconnection differences. Yet to the best of our knowledge a formal investigation of model covariances and the prior dependency of PaleoDA has yet to be done. We hope that our results can initiate further resarch into that important topic alongside the mentioned studies.

We agree, that the lack of signal is usually sign of a noisy reconstruction, for instance for the tree-only reconstruction. In fact, this is also visible in the uncertainties of our reconstructed monsoon indices based on different proxy data (two standard deviations of the reconstructed ensemble), which are we not showing to keep the figures clearer. The reconstructions uncertainties are in general quite large, as is usually the case with the PaleoDA method. We will mention the noisiness of the signals with smaller variability in the revised manuscript. The reconstruction uncertainty is also a clear indicator of how much a reconstruction has been affected by the proxy records. We will include an appendix figure of the reconstruction uncertainty to include this important aspect in our study.

Action: Done. Figure 6 for the single model prior precipitation reconstruction was added to the main text, and the prior dependency mentioned in the Results section (line 478ff) and in the Discussion section 6.1 (553ff), including some further suggestions how the influence of the different prior models could be studied:

"While the magnitude of the wet conditions also proved to be model prior dependent, the source of model differences in the reconstruction cannot be directly deduced from the simulated SASM indices and the correlations in the model, requiring further analysis of the topic. In addition to the correlation analysis that we did for the core monsoon region and NEB, an optimal sensor placement analysis as performed by Comboul et al. (2015)

and King et al. (2023) could give insights into how different proxy records influence the reconstruction depending on the model prior."

Action: Figure 7 for the uncertainty was added to the Appendix, and mentioned in the main text (line 466ff):

"Furthermore, the smaller reconstruction uncertainty for reconstructions involving speleothems is noticeable, especially compared to the tree ring only reconstruction (AF D6), though the overall uncertainty due to the large spread in the prior ensemble remains large."

[Figure]

Figure 6: Single model reconstructions of the monsoon precipitation index using all proxy records, highlighting the prior dependency of the precipitation reconstruction. The black denotes the the multi-model mean, which is used in the multi-model ensemble analysis.

L450: I think it should be AF11 not AF12.
Thank you for pointing out this typo.

Action: Done.

**2 Reply to the second reviewer**

*This manuscript presents a new Common Era climate reconstruction of South America, generated via paleo data assimilation. The authors include speleothems in their methodology, a largely unused archive in paleo data reconstructions. The manuscript provides a detailed description of their DA methods, validation methods, and clearly discuss the methodological limitations. They use their climate reconstructions to examine climatological anomalies through time and compare their findings to existing DA products.*

[Figure]

Figure 7: Reconstruction uncertainty for the indices from Figure 4 defined as the standard deviation of the posterior ensemble. Here, we display the mean of the standard deviations of the five single model reconstructions, although also other multi-model ensemble error definitions in terms of the propagation of uncertainty are conceivable.

*This manuscript was incredibly thorough and presents an exciting new advancement in the use of speleothem records in paleodata assimilation. I think the authors hit a good balance between interpreting their reconstruction and an honest discussion of the limitations/uncertainties. From my understanding of DA, the methodology and science seems sounds, and I believe the manuscript is nearly ready for publication. I focus my few comments here on organization and readability.*

We thank the reviewer for this positive assessment.

*Appendix organization: In my opinion, I think the Appendices are out of order (e.g. appendix D is reference before appendix C). This makes navigating through the very lengthy supplemental information a bit challenging. I would recommend reordering the appendices as follows (A, D, C, B). I ran into a similar issue with the supplemental figures within Appendix A. There were several times where figures were referenced out of order (e.g. AF9 and 10 were referenced before AF 5-8).*

Thank you for pointing this out. We agree with the reviewer, that the Appendices are out of order and need to be in line with order of appearance in the main text. Therefore we will reorder and reorganize the Appendices, but will slightly deviate from the order suggested by the reviewer (D,C,B,A):

- **D** The data tables of the employed proxy records will be mentioned first. We will also include the additional figure for the proxy record distribution (currently AF1) in that section. The former section A will become a section that includes only additional figures concerning the results.

- **C** This section will be changed into an Appendix Section for the Methodology. It will include the algorithm sketch (currently AF 2), a new sketch for the multi-time scale approach which was suggested by Reviewer 1 and the derivation of the SNR based proxy record error.

- **B** The validation of the reconstruction with instrumental data appears before the actual results in the main text.

- **A** for the additional figures of the results section.

- The Appendix names will be renamed to match the alphabetical order.

After the adjustment of the appendices, we will check the order of appearances of figures in the main text. We think that the reorganization of the appendices, for instance only

including figures regarding the results in appendix A (new appendix D), will make the order of appearance less confusing.

Action: Done. The order of appearances of all Appendix Figures has been checked. Note, that the consistency can only be guaranteed for the first appearance, as Figures are sometimes refered to again at a later point in the text.

*Supplemental figures: The number of supplemental figures in Appendix A makes the manuscript a bit cumbersome to read. I also find that the authors extensively discuss many of their supplemental figures in the text. This makes me wonder whether some should be moved to the main text? While I acknowledge that this may make the flow a bit less elegant, I think it would help guide the reader as the manuscript touches on quite a few different topics. I will not recommend any specific changes here and leave this decision up to the authors.*

Thank you for appreciating our supplement figures and for leaving us the freedom to redesign the importance. We will move figure AF12, which shows the prior model dependency of the reconstruction, but only the precipitation part (a), into the main text. We consider the prior dependency of the precipitation reconstruction an important aspect of our study which should be highlighted more, as also Reviewer 1 has pronounced interest about this topic. We hope that the reorganization of the appendix will also help in making the appendix less cumbersome to read.

Action: Done.

**Detailed Comments**

*Line 65 – Missing some detail on the drivers of speleothem d18O. E.g. are there any studies specific to South America that should be cited here? What about upstream rainout, cloud effects. Etc... Dansgaard is a good reference, but some additional works should be cited here.*

Thank you. We will be more specific to South America and change the section as follows:

"Speleothems are geological cave formations created by accumulating layers of calcium carbonates transported by seepage water. Among the many climate proxies archived in speleothems, the ratio between heavy and light oxygen isotopes ($\delta^{18}O$) as saved in accumulating layers of calcium carbonate reflects the isotopic composition of the precipitation above a cave and, thus, records hydroclimatic changes (Bradley, 2015). The $\delta^{18}O$ signatures of precipitation are sensitive to air temperature, precipitation amount changes, and the geographical location in terms of altitude, latitude, and distance from the coast (Dansgaard, 1964). **For South America, in particular the SASM influenced region, the rainfall amount during the monsoon season is a primary driver on the $\delta^{18}O$ signatures of precipitation (Vuille et al., 2003; Moquet et al., 2016)"**

Action: Done.

*Line 75 – topic sentence is a bit confusing – perhaps say 'are excluded' instead of 'may be excluded'*

Thank you for pointing this out. 'may be excluded' was used to indicate that the insights from speleothem proxy based studies of South American climate during the CE could be missing in climate field reconstructions if not conveyed by other types of proxy records from South America. We will change the sentence as follows to make it clearer.
"...**It is not clear, if existing climate field reconstructions include these insights into South American Hydroclimate variability during the CE due to the limited integration of speleothem records.** ..."

Action: Done.

*Line 399 – Unless I missed it, I couldn't find the definition of the Southern Cone*
Thank you for pointing this our. We will add a short definition, where it is first mentioned in line 167 as follows:
"...Regions lacking archive sites for proxy records can be found in the northern part of South America, namely Colombia, the Guianas and the north western states of Brazil. Additionally, the western part of the Southern Cone**, the cone-shaped area of South America south of the Tropic of Capricorn ($\sim$23.4°S),** lacks proxy records. However, the South American Drought Atlas has demonstrated that tree ring records from the central and southern Andes can be skillfully used to reconstruct the hydroclimate of that region...."

Action: Done.

**References**

Boës, X. and Fagel, N.: Relationships between southern Chilean varved lake sediments, precipitation and ENSO for the last 600 years, Journal of Paleolimnology, 39, 237–252, https://doi.org/10.1007/s10933-007-9119-9, 2007.

Bradley, R.: Paleoclimatology. Reconstructing Climates of the Quaternary., Elsevier, 2015.

Bühler, J. C., Axelsson, J., Lechleitner, F. A., Fohlmeister, J., LeGrande, A. N., Midhun, M., Sjolte, J., Werner, M., Yoshimura, K., and Rehfeld, K.: Investigating Stable Oxygen and Carbon Isotopic Variability in Speleothem Records over the Last Millennium

Using Multiple Isotope-Enabled Climate Models, Climate of the Past, 18, 1625–1654, https://doi.org/10.5194/cp-18-1625-2022, 2022.

Burgdorf, A.-M., Broennimann, S., Adamson, G., Amano, T., Aono, Y., Barriopedro, D., Bullon, T., Camenisch, C., Camuffo, D., Daux, V., del Rosario Prieto, M., Dobrovolny, P., Gallego, D., Garcia-Herrera, R., Gergis, J., Grab, S., Hannaford, M. J., Holopainen, J., Kelso, C., Kern, Z., Kiss, A., Kuan-Hui Lin, E., Loader, N. J., Mozny, M., Nash, D., Nicholson, S. E., Pfister, C., Rodrigo, F. S., Rutishauser, T., Sharma, S., Takacs, K., Vargas, E. T., and Vega, I.: DOCU-CLIM: A global documentary climate dataset for climate reconstructions, Scientific Data, 10, https://doi.org/10.1038/s41597-023-02303-y, 2023.

Comboul, M., Emile-Geay, J., Hakim, G. J., and Evans, M. N.: Paleoclimate Sampling as a Sensor Placement Problem, Journal of Climate, 28, 7717–7740, https://doi.org/10.1175/JCLI-D-14-00802.1, 2015.

Compo, G. P., Whitaker, J. S., Sardeshmukh, P. D., Matsui, N., Allan, R. J., Yin, X., Gleason, B. E., Vose, R. S., Rutledge, G., Bessemoulin, P., Brönnimann, S., Brunet, M., Crouthamel, R. I., Grant, A. N., Groisman, P. Y., Jones, P. D., Kruk, M. C., Kruger, A. C., Marshall, G. J., Maugeri, M., Mok, H. Y., Nordli, Ø., Ross, T. F., Trigo, R. M., Wang, X. L., Woodruff, S. D., and Worley, S. J.: The Twentieth Century Reanalysis Project, Quarterly Journal of the Royal Meteorological Society, 137, 1–28, https://doi.org/10.1002/qj.776, 2011.

Conroy, J. L., Restrepo, A., Overpeck, J. T., Steinitz-Kannan, M., Cole, J. E., Bush, M. B., and Colinvaux, P. A.: Unprecedented recent warming of surface temperatures in the eastern tropical Pacific Ocean, Nature Geoscience, 2, 46–50, https://doi.org/10.1038/ngeo390, 2008.

Cook, E. R., Meko, D. M., Stahle, D. W., and Cleaveland, M. K.: Drought Reconstructions for the Continental United States*, Journal of Climate, 12, 1145–1162, https://doi.org/10.1175/1520-0442(1999)012⟨1145:drftcu⟩2.0.co;2, 1999.

Dansgaard, W.: Stable Isotopes in Precipitation, Tellus, 16, 436–468, https://doi.org/10.1111/j.2153-3490.1964.tb00181.x, 1964.

Dee, S. G., Steiger, N. J., Emile-Geay, J., and Hakim, G. J.: On the utility of proxy system models for estimating climate states over the common era, Journal of Advances in Modeling Earth Systems, 8, 1164–1179, https://doi.org/10.1002/2016ms000677, 2016.

Haberzettl, T., Fey, M., Lücke, A., Maidana, N., Mayr, C., Ohlendorf, C., Schäbitz, F., Schleser, G. H., Wille, M., and Zolitschka, B.: Climatically induced lake level changes during the last two millennia as reflected in sediments of Laguna Potrok Aike,

southern Patagonia (Santa Cruz, Argentina), Journal of Paleolimnology, 33, 283–302, https://doi.org/10.1007/s10933-004-5331-z, 2005.

Hakim, G. J., Emile-Geay, J., Steig, E. J., Noone, D., Anderson, D. M., Tardif, R., Steiger, N., and Perkins, W. A.: The Last Millennium Climate Reanalysis Project: Framework and First Results, Journal of Geophysical Research: Atmospheres, 121, 6745–6764, https://doi.org/10.1002/2016JD024751, 2016.

Harris, I., Osborn, T. J., Jones, P., and Lister, D.: Version 4 of the CRU TS monthly high-resolution gridded multivariate climate dataset, Scientific Data, 7, https://doi.org/10.1038/s41597-020-0453-3, 2020.

Jiménez-Iñiguez, A., Ampuero, A., Valencia, B. G., Mayta, V. C., Cruz, F. W., Vuille, M., Novello, V. F., Misailidis Stríkis, N., Aranda, N., and Conicelli, B.: Stable isotope variability of precipitation and cave drip-water at Jumandy cave, western Amazon River basin (Ecuador), J. Hydrol. (Amst.), 610, 127 848, 2022.

King, J., Anchukaitis, K. J., Allen, K., Vance, T., and Hessl, A.: Trends and Variability in the Southern Annular Mode over the Common Era, Nature Communications, 14, 2324, https://doi.org/10.1038/s41467-023-37643-1, 2023.

King, J. M., Anchukaitis, K. J., Tierney, J. E., Hakim, G. J., Emile-Geay, J., Zhu, F., and Wilson, R.: A Data Assimilation Approach to Last Millennium Temperature Field Reconstruction Using a Limited High-Sensitivity Proxy Network, Journal of Climate, 34, 7091–7111, https://doi.org/10.1175/JCLI-D-20-0661.1, 2021.

Moquet, J. S., Cruz, F. W., Novello, V. F., Stríkis, N. M., Deininger, M., Karmann, I., Santos, R. V., Millo, C., Apaestegui, J., Guyot, J. L., Siffedine, A., Vuille, M., Cheng, H., Edwards, R. L., and Santini, W.: Calibration of Speleothem $\delta 18O$ Records against Hydroclimate Instrumental Records in Central Brazil, Global and Planetary Change, 139, 151–164, https://doi.org/10.1016/j.gloplacha.2016.02.001, 2016.

Morales, M. S., Cook, E. R., Barichivich, J., Christie, D. A., Villalba, R., LeQuesne, C., Srur, A. M., Ferrero, M. E., González-Reyes, Á., Couvreux, F., et al.: Six hundred years of South American tree rings reveal an increase in severe hydroclimatic events since mid-20th century, Proceedings of the National Academy of Sciences, 117, 16 816–16 823, 2020.

Neukom, R., del Rosario Prieto, M., Moyano, R., Luterbacher, J., Pfister, C., Villalba, R., Jones, P. D., and Wanner, H.: An Extended Network of Documentary Data from South America and Its Potential for Quantitative Precipitation Reconstructions Back to the 16th Century, Geophysical Research Letters, 36, L12 703, https://doi.org/10.1029/2009GL038351, 2009.

Neukom, R., Gergis, J., Karoly, D., Wanner, H., Curran, M., Elbert, J., González Rouco, J. F., Linsley, B., Moy, A., Mundo, I., Raible, C., Steig, E., van Ommen, T., Vance, T., Villalba, R., Zinke, J., and Frank, D.: Inter-Hemispheric Temperature Variability over the Last Millennium, Nature Climate Change, 4, https://doi.org/10.1038/nclimate2174, 2014.

Orrison, R., Vuille, M., Smerdon, J. E., Apaéstegui, J., Azevedo, V., Campos, J. L. P. S., Cruz, F. W., Della Libera, M. E., and Stríkis, N. M.: South American Summer Monsoon Variability over the Last Millennium in Paleoclimate Records and Isotope-Enabled Climate Models, Climate of the Past, 18, 2045–2062, https://doi.org/10.5194/cp-18-2045-2022, 2022.

Parsons, L. A., Amrhein, D. E., Sanchez, S. C., Tardif, R., Brennan, M. K., and Hakim, G. J.: Do Multi-Model Ensembles Improve Reconstruction Skill in Paleoclimate Data Assimilation?, Earth and Space Science, 8, e2020EA001 467, https://doi.org/10.1029/2020EA001467, 2021.

Rein, B.: How do the 1982/83 and 1997/98 El Niños rank in a geological record from Peru?, Quaternary International, 161, 56–66, https://doi.org/10.1016/j.quaint.2006.10.023, 2007.

Sanchez, S. C., Hakim, G. J., and Saenger, C. P.: Climate Model Teleconnection Patterns Govern the Niño-3.4 Response to Early Nineteenth-Century Volcanism in Coral-Based Data Assimilation Reconstructions, Journal of Climate, 34, 1863–1880, https://doi.org/10.1175/JCLI-D-20-0549.1, 2021.

Schneider, U., Fuchs, T., Meyer-Christoffer, A., and Rudolf, B.: Global precipitation analysis products of the GPCC, Global Precipitation Climatology Centre (GPCC), DWD, Internet Publikation, 112, 2008.

Smerdon, J. E.: Climate Models as a Test Bed for Climate Reconstruction Methods: Pseudoproxy Experiments: Pseudoproxy Experiments, Wiley Interdisciplinary Reviews: Climate Change, 3, 63–77, https://doi.org/10.1002/wcc.149, 2012.

Steiger, N. J., Smerdon, J. E., Cook, E. R., and Cook, B. I.: A Reconstruction of Global Hydroclimate and Dynamical Variables over the Common Era, Scientific Data, 5, 180 086, https://doi.org/10.1038/sdata.2018.86, 2018.

Tardif, R., Hakim, G. J., Perkins, W. A., Horlick, K. A., Erb, M. P., Emile-Geay, J., Anderson, D. M., Steig, E. J., and Noone, D.: Last Millennium Reanalysis with an Expanded Proxy Database and Seasonal Proxy Modeling, Climate of the Past, 15, 1251–1273, https://doi.org/10.5194/cp-15-1251-2019, 2019.

Valler, V., Franke, J., Brugnara, Y., Samakinwa, E., Hand, R., Lundstad, E., Burgdorf, A.-M., Lipfert, L., Friedman, A. R., and Brönnimann, S.: ModE-RA: a global monthly paleo-reanalysis of the modern era 1421 to 2008, Scientific Data, 11, https://doi.org/10.1038/s41597-023-02733-8, 2024.

Vuille, M., Bradley, R. S., Werner, M., Healy, R., and Keimig, F.: Modeling d18O in precipitation over the tropical Americas: 1. Interannual variability and climatic controls, Journal of Geophysical Research: Atmospheres, 108, https://doi.org/10.1029/2001jd002038, 2003.

Wang, J., Emile-Geay, J., Guillot, D., Smerdon, J. E., and Rajaratnam, B.: Evaluating Climate Field Reconstruction Techniques Using Improved Emulations of Real-World Conditions, Climate of the Past, 10, 1–19, https://doi.org/10.5194/cp-10-1-2014, 2014.

---

## Author Response (AR2)

**Summary of changes for revised manuscript: A continental reconstruction of hydroclimatic variability in South America during the past 2000 years (egusphere-2024-545)**

Mathurin A. Choblet, Janica C. Bühler, Valdir F. Novello, Nathan J. Steiger, and Kira Rehfeld

August 8, 2024

**Summary of changes (minor revision)**

Dear Kathleen Wendt,

Thank you for the positive assessment of the manuscript revision.

As suggested, we have further expanded the discussion of the drivers influencing $\delta^{18}$O in precipitation:

**For South America, particularly in the South American Summer Monsoon (SASM) influenced region, the primary driver of $\delta^{18}O$ signatures in precipitation is the amount of rainfall during the monsoon season, rather than temperature (Vuille et al., 2003; Moquet et al., 2016). Additionally, the $\delta^{18}O$ signatures are influenced by the location of the moisture source, notably the moisture contribution from the ITCZ region, and the degree of upstream rainout, which captures the precipitation history of the air mass along its trajectory as demonstrated by cave monitoring studies (Ampuero et al., 2020; Jiménez-Iñiguez et al., 2022; Moquet et al., 2016) and modeling studies (Vuille et al., 2003; Vuille and Werner, 2005; Vuille et al., 2012) working with station data from the Global Network of Isotopes in Precipitation (GNIP) (IAEA/WMO, 2020). $\delta^{18}O$ in speleothems in tropical South America is thus closely linked to changes in both regional and large-scale atmospheric circulation patterns.**

Additionally, the first two sentences of the abstract has been reworded to emphasize the importance of understanding hydroclimatic variability:

**Paleoclimatological field reconstructions are valuable for understanding past hydroclimatic variability, which is crucial for assessing potential future hydroclimate changes. Despite being as impactful on societies as temperature variability, hydroclimatic variability—particularly beyond the instrumental record—has received less attention.**

We have also adjusted the numbering and naming of the supplement sections and checked that the color cycles for the line plots show a reasonable contrast under different colorblind filters.

Best regards,

Mathurin Choblet
On behalf of the authors

**References**

Ampuero, A., Stríkis, N. M., Apaéstegui, J., Vuille, M., Novello, V. F., Espinoza, J. C., Cruz, F. W., Vonhof, H., Mayta, V. C., Martins, V. T. S., Cordeiro, R. C., Azevedo, V., and Sifeddine, A.: The Forest Effects on the Isotopic Composition of Rainfall in the Northwestern Amazon Basin, Journal of Geophysical Research: Atmospheres, 125, https://doi.org/10.1029/2019jd031445, 2020.

IAEA/WMO: Global Network of Isotopes in Precipitation. The GNIP Database, URL `http://www.iaea.org/water`, accessible at: `http://www.iaea.org/water`, 2020.

Jiménez-Iñiguez, A., Ampuero, A., Valencia, B. G., Mayta, V. C., Cruz, F. W., Vuille, M., Novello, V. F., Misailidis Stríkis, N., Aranda, N., and Conicelli, B.: Stable isotope variability of precipitation and cave drip-water at Jumandy cave, western Amazon River basin (Ecuador), J. Hydrol. (Amst.), 610, 127 848, 2022.

Moquet, J. S., Cruz, F. W., Novello, V. F., Stríkis, N. M., Deininger, M., Karmann, I., Santos, R. V., Millo, C., Apaestegui, J., Guyot, J. L., Siffedine, A., Vuille, M., Cheng, H., Edwards, R. L., and Santini, W.: Calibration of Speleothem $\delta$18O Records against Hydroclimate Instrumental Records in Central Brazil, Global and Planetary Change, 139, 151–164, https://doi.org/10.1016/j.gloplacha.2016.02.001, 2016.

Vuille, M. and Werner, M.: Stable isotopes in precipitation recording South American summer monsoon and ENSO variability: observations and model results, Climate Dynamics, 25, 401–413, https://doi.org/10.1007/s00382-005-0049-9, 2005.

Vuille, M., Bradley, R. S., Werner, M., Healy, R., and Keimig, F.: Modeling d18O in precipitation over the tropical Americas: 1. Interannual variability and climatic controls, Journal of Geophysical Research: Atmospheres, 108, https://doi.org/10.1029/2001jd002038, 2003.

Vuille, M., Burns, S. J., Taylor, B. L., Cruz, F. W., Bird, B. W., Abbott, M. B., Kanner, L. C., Cheng, H., and Novello, V. F.: A Review of the South American Monsoon History as Recorded in Stable Isotopic Proxies over the Past Two Millennia, Climate of the Past, 8, 1309–1321, https://doi.org/10.5194/cp-8-1309-2012, 2012.